# Signals from the brain and olfactory epithelium control shaping of the mammalian nasal capsule cartilage

Marketa Kaucka[1,2†], Julian Petersen[1,2†], Marketa Tesarova[3], Bara Szarowska[2], Maria Eleni Kastriti[1,2], Meng Xie[1], Anna Kicheva[4], Karl Annusver[5,6], Maria Kasper[5,6], Orsolya Symmons[7], Leslie Pan[8], Francois Spitz[8,9], Jozef Kaiser[3], Maria Hovorakova[10], Tomas Zikmund[3], Kazunori Sunadome[1], Michael P Matise[11], Hui Wang[11], Ulrika Marklund[12], Hind Abdo[12], Patrik Ernfors[12], Pascal Maire[13], Maud Wurmser[13], Andrei S Chagin[1,14], Kaj Fried[15], Igor Adameyko[1,2*]

[1]Department of Physiology and Pharmacology, Karolinska Institutet, Stockholm, Sweden; [2]Department of Molecular Neurosciences, Medical University Vienna, Vienna, Austria; [3]Central European Institute of Technology, Brno University of Technology, Brno, Czech Republic; [4]Institute of Science and Technology IST Austria, Klosterneuburg, Austria; [5]Department of Biosciences and Nutrition, Karolinska Institutet, Stockholm, Sweden; [6]Center for Innovative Medicine, Karolinska Institutet, Huddinge, Sweden; [7]Department of Bioengineering, University of Pennsylvania, Philadelphia, United States; [8]Developmental Biology Unit, European Molecular Biology Laboratory, Heidelberg, Germany; [9]Genomics of Animal Development Unit, Institut Pasteur, Paris, France; [10]Department of Developmental Biology, Institute of Experimental Medicine, The Czech Academy of Sciences, Prague, Czech Republic; [11]Department of Neuroscience & Cell Biology, Rutgers-Robert Wood Johnson Medical School, Piscataway, United States; [12]Department of Medical Biochemistry and Biophysics, Karolinska Institutet, Stockholm, Sweden; [13]Department of Development, Reproduction and Cancer, Institute Cochin, Paris, France; [14]Institute for Regenerative Medicine, Sechenov First Moscow State Medical University, Moscow, Russia; [15]Department of Neuroscience, Karolinska Institutet, Stockholm, Sweden

*For correspondence: igor.adameyko@ki.se

†These authors contributed equally to this work

Competing interests: The authors declare that no competing interests exist.

**Abstract** Facial shape is the basis for facial recognition and categorization. Facial features reflect the underlying geometry of the skeletal structures. Here, we reveal that cartilaginous nasal capsule (corresponding to upper jaw and face) is shaped by signals generated by neural structures: brain and olfactory epithelium. Brain-derived Sonic Hedgehog (SHH) enables the induction of nasal septum and posterior nasal capsule, whereas the formation of a capsule roof is controlled by signals from the olfactory epithelium. Unexpectedly, the cartilage of the nasal capsule turned out to be important for shaping membranous facial bones during development. This suggests that conserved neurosensory structures could benefit from protection and have evolved signals inducing cranial cartilages encasing them. Experiments with mutant mice revealed that the genomic regulatory regions controlling production of SHH in the nervous system contribute to facial cartilage morphogenesis, which might be a mechanism responsible for the adaptive evolution of animal faces and snouts.
DOI: https://doi.org/10.7554/eLife.34465.001

## Introduction

The shape of a face strongly depends on the geometry of skeletal elements directly under the skin, adipose tissue and muscles. Our adult cranial and, in particular, facial skeleton consists mostly of bony elements. Cartilaginous parts are rather minor. However, during embryonic development bone forms after the cartilage, and the initial phases of facial and skull shaping proceed with the cartilaginous skeleton only. The entire functional and evolutionary meaning of the chondrocranium, that is, the early cartilaginous elements of the embryonic skull, is not clear. Some parts of the chondrocranium will undergo endochondral ossification (for example, pre-sphenoid and basisphenoid, cribriform plate, Meckel´s cartilage, olfactory septum, nasal concha, labyrinth of ethmoid, vomer and tympanic bulla). However, the majority of the bones, especially in a facial compartment, will form in a close spatial association with the chondrocranium independently through dermal membranous ossifications (*Carson, 1999*). Many questions, including how and from where molecular signals control the complex chondrocranial shape, and whether the geometry of the chondrocranium directs the shape of facial bones, are still unanswered.

Achondroplasia, a rare disease due to cartilage insufficiency, includes craniofacial malformations such as protruding forehead, low nasal bridge, maxillary hypoplasia, problems in the otolaryngeal system and macrocephaly as well as foramen magnum stenosis (*Shirley and Ain, 2009*). Prominent human and mouse achondroplasia phenotypes based on *Fgfr3* mutations suggest that a correctly shaped chondrocranium is essential for proper facial bone geometry and general facial outgrowth. However, *Fgfr3* is expressed also at membranous ossification sites (please see *Fgfr3*(mRNA) expression at E15, http://developingmouse.brain-map.org/), as well as in sutural osteogenic fronts (*Ornitz and Marie, 2002*). Therefore, models involving *Fgfr3* do not allow for precise discrimination of cartilage or bone-dependent parts of the phenotype in affected subjects. Still, these effects strongly suggest that chondrocranium shape might be truly important for producing initial facial geometry and for influencing the formation of membranous bone on top of the cartilaginous template.

The facial chondrocranium is built by neural crest cells that populate the frontal part of the head and undergo multilineage differentiation. They give rise to cartilage, bone, fascia, adipose tissue, smooth muscle, pericytes, glia and neurons (*Snider and Mishina, 2014*; *Baggiolini et al., 2015*). Paraxial mesoderm also contributes to the chondrocranium in posterior basicranial and occipital locations. Collective behavior and differentiation of the neural crest and neural crest-derived ectomesenchyme is largely responsible for the future shape of the face (*Minoux and Rijli, 2010*). However, the precise mechanisms governing this collective behavior, cartilage induction and shape-making are not fully understood, despite significant progress in the research field of cartilage and bone formation.

McBratney-Owen and Morris-Key with coworkers demonstrated that the complete chondrocranium (including the base of the skull) develops from 14 pairs of independently induced large cartilaginous elements that fuse together during later development (*McBratney-Owen et al., 2008*). Sculpting perfect geometries of such cartilaginous elements is a key developmental and regenerative process that accounts for the shape and integrity of our body. Current opinion holds that cartilage forms from condensing mesenchymal cells that are destined to become chondrocytes (*Ornitz and Marie, 2002*). Mesenchymal condensations emerge in specific locations. Here, they somehow become shaped, grow and turn into cartilage that later expands until the initiation of endochondral or membranous ossification.

The frontonasal prominence and other facial regions are enriched in signaling systems. Activity in these systems leads to progressive induction and shaping of craniofacial structures, including chondrogenic mesenchymal condensations that turn into cartilage (*Minoux and Rijli, 2010*). The signaling center located in the most anterior face, the so called FEZ (Frontonasal Ectodermal Zone), produces Sonic Hedgehog (SHH) and Fibroblast growth factor 8 (FGF8), which play important roles in facial shaping. FGF8, SHH and Bone Morphogenetic Proteins (BMPs) produced by FEZ regulate the behavior of ectomesenchymal tissue and participate in positioning of chondrogenic condensations inside of the embryonic face (*Foppiano et al., 2007*; *Hu et al., 2015b*; *Young et al., 2014*). The mechanisms of facial cartilage induction that involve these molecules have received particular attention during recent years (*Gros and Tabin, 2014*; *Abzhanov and Tabin, 2004*; *Bhullar et al., 2015*; *Griffin et al., 2013*).

Another recent breakthrough brought up the fact that the brain itself can emit signals that influence facial shaping. Expression of *Shh* in the forebrain turned out to be important for the correct formation of FEZ and early steps of facial shaping in general (*Hu et al., 2015a*; *Chong et al., 2012*).

Still, how these and other signaling centers synchronize in order to build the 3D shape of facial cartilaginous elements is not understood. The cartilaginous nasal capsule is the most anterior part of the chondrocranium. Together with Meckel's cartilage in the lower jaw, it constitutes an excellent model system to address questions concerning cartilage induction and shaping.

Here, using mild ablations of cartilage with tightly controlled genetic tools, we revealed that the shape of the nasal capsule is a key for the geometry and positioning of the facial bones and overall facial shape. Subsequently, with the help of numerous mouse mutants, specific contrasting techniques and micro-CT, we demonstrated that signaling centers in the developing brain and olfactory epithelium jointly and independently enable the induction of the nasal capsule in the embryonic face. Various genomic regulatory regions that direct the expression of *Shh* to the developing nervous system participate in the fine-tuning of the shape of the facial cartilaginous skeleton.

## Results

Taking into account known achondroplasia facial phenotypes, we hypothesized that even minor changes in the facial cartilaginous template may lead to significant or even major changes in the overlaying membranous bone geometry and the overall facial shape. Thus, we performed mild time-controlled genetic ablation of early chondrocytes employing *Col2a1-CreERT2/R26DTA* mice and analyzed their facial development (*Figure 1*). We used a dose of 2.5 mg of tamoxifen administered at E12.5 and, in an alternative experiment, double injection at E13.5-E14.5 to avoid a strong phenotype with dramatic face shortening and brain shape distortion, and we analyzed the mutant embryos at E17.5 and E15.5 correspondingly (*Figure 1*). *Col2a1-CreERT2* is a well-established tool to target facial chondrocytes and their immediate progenitors. Tamoxifen was administered at early stages of facial development where no bone is present. Also, this *CreERT2* line does not recombine in osteoblasts or their progenitors in membranous ossification sites and, thus, cannot directly impinge on them (*Figure 1A–D*). Despite only mild cartilage reduction (mean 30,7% of the cartilage surface decrease at E15.5 and mean 35,2% at E17.5), the facial compartment of the *Col2a1-CreERT2$^{+/-}$/R26DTA$^{+/-}$*embryos appeared massively affected with short snout and distorted membranous ossifications (*Figure 1E–R*). Interestingly, the forming mandibular bone appeared shortened and widened at the same time, which cannot be explained only by the shortening of Meckel's cartilage (*Figure 1—figure supplement 1*). This fact suggests an interplay between cartilage and membranous bone that might involve signal-guided rearrangements in skeletogenic tissues. Thus, genetic ablations of COL2A1-producing pre-chondrocytes and chondrocytes revealed a different degree of cartilage loss in the nasal capsule and Meckel's cartilage, together with corresponding incremental dysmorphologies of membranous bones and face in general. These slight differences in the strength of the phenotype are likely attributed to the fine diversity of developmental stages within one litter receiving tamoxifen during a single injection into a mother (*Figure 1Q–R* and *Figure 1—figure supplement 1*). Thus, the geometry of the face and corresponding bony structures depend on the correct induction and shaping of a facial cartilage. This, in turn, is largely established at the level of chondrogenic mesenchymal condensations, as we recently demonstrated (*Kaucka et al., 2017*). It is worth emphasizing that according to our previous study, the chondrogenic condensations are induced being pre-shaped (*Kaucka et al., 2017*).

Consequently, we decided to analyze the molecular signals and their sources that induce these geometrically complex condensations. Several molecules were reported to have an impact on the cartilage induction, either on the condensation placement or on proper timing of cartilage-forming events (*Goldring et al., 2006*). Among those, SHH was shown to play a key role in the spatio-temporal induction of chondrogenic mesenchymal condensations (*Abzhanov and Tabin, 2004*; *Billmyre and Klingensmith, 2015*; *Park et al., 2010*). We analyzed the expression of *Shh* in early and late developing mouse face with the help of the *Shh-Cre/GFP (B6.Cg-Shhtm1(EGFP/cre)Cjt/J)* model, and found that *Shh* is expressed in very discrete regions of the face between E11.5 and E14.5 at the time of induction of facial cartilages (*Figure 2* and *Figure 2—figure supplement 1*). At the early stages (E11.5 and E12.5, see *Figure 2A–B*), the SHH + regions included forming olfactory epithelium (magenta), dental and oral epithelium (red), eyes (cyan) and brain (yellow). Interestingly,

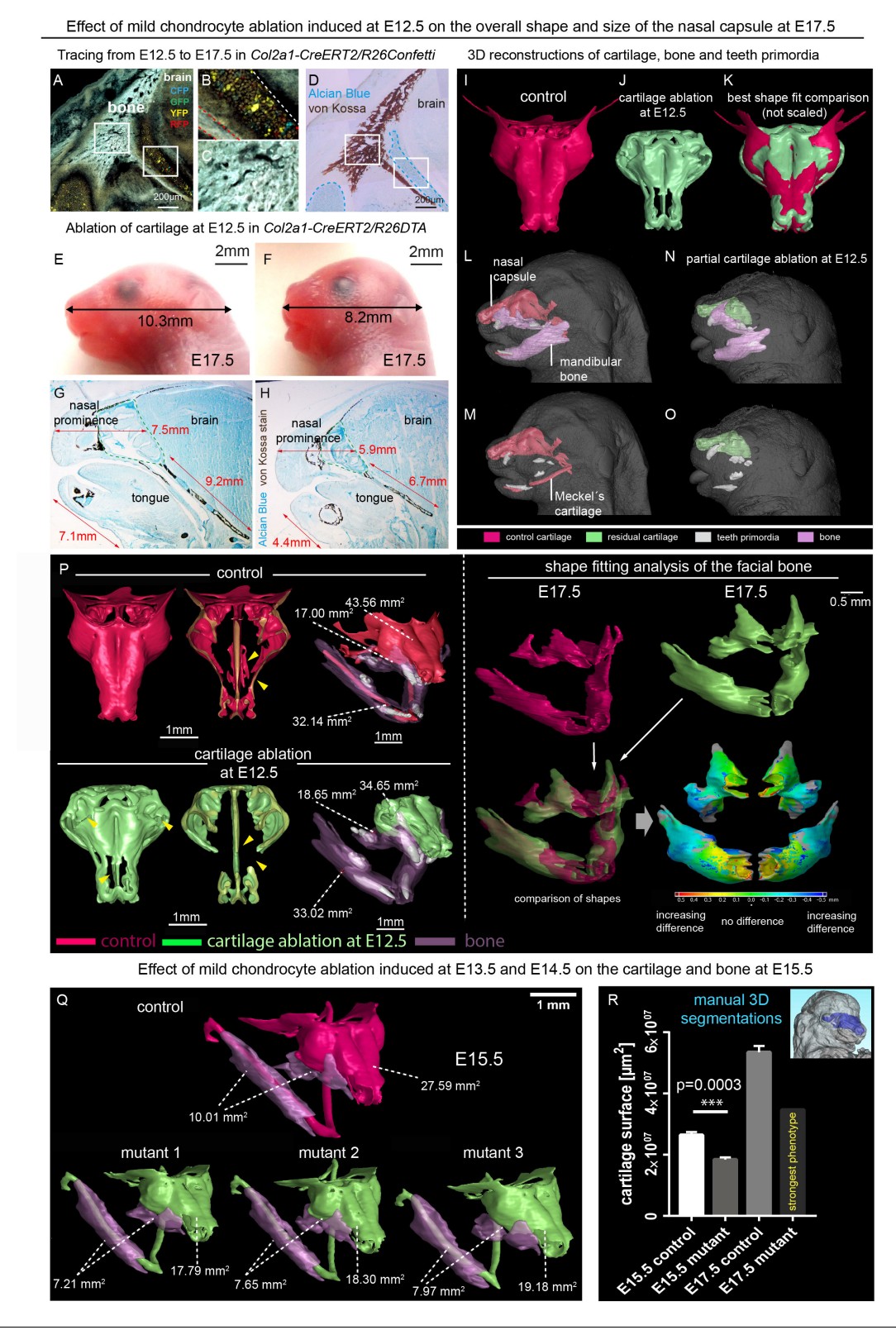

**Figure 1.** Correct chondrocranium development is essential for shaping the embryonic face. (**A–C**) Genetic tracing induced at E12.5 in *Col2a1-CreERT2/R26Confetti* shows recombination in chondrocytes (**B**) only and not in a lineage of membranous bone osteoblasts and their progenitors (**C**), 25 µm cryo-sections (**A–D, G–H**) were imaged with a confocal microscope (**A–C**) or phase contrast microscope (**D, G–H**). (**D**) Traced sections have been stained using Alcian Blue (cartilage, blue) and von Kossa (brown, mineralized tissue). (**E–F**) Wild type (**E**) and *Col2a1-CreERT2/R26DTA* (**F**) embryos with *Figure 1 continued on next page*

*Figure 1 continued*

cartilage being partially ablated as a result of tamoxifen injection (2.5 mg) at E12.5, both analyzed at E17.5. (G–H) Sagittal sections of the anterior head from wild type (G) and *Col2a1-CreERT2/R26DTA* (H) embryos stained with Alcian Blue (blue, stains for cartilage) and von Kossa staining (black, stains for mineralized bone tissue). Olfactory system is outlined by green dashed line for better orientation. Note that physiological growth of the cartilage sets the proper size of the facial compartment. (I–O) 3D-reconstructions of frontal chondrocranium together with bone and teeth primordia in control (I, L, M) and cartilage-ablated (J, N, O) embryos. (K) Best fit comparison of control (red) and cartilage-ablated (light green) 3D chondrocranium models. (P) 3D-reconstruction of frontal chondrocranium and formed cartilage including GOM Inspect software analysis of the mutant bone (Q) overview of analyzed mutants (injected with low dose of tamoxifen (2.5 mg) at both E13.5 and E14.5 and analyzed at E15.5) and formed bone in one representative control and three mutants (R) Bar-graphs showing the manual 3D segmentation of the surface area of cartilage. Data are obtained from three control samples and three mutant mice for (E15.5) and three control samples and one mutant sample with the most pronounced phenotype for (E17.5). The error bars show mean and standard deviation (SD). For the comparison, we used unpaired Student t-test (95% confidence interval −9974138 to −6056665). Raw data are available in *Figure 1—source data 1*.
DOI: https://doi.org/10.7554/eLife.34465.002

The following source data and figure supplement are available for figure 1:

**Source data 1.** Raw values of cartilage surface measurments corresponding to Graph in *Figure 1R*.
DOI: https://doi.org/10.7554/eLife.34465.004

**Figure supplement 1.** Mild ablation of cartilage using *Col2a1-CreERT2/R26DTA*.
DOI: https://doi.org/10.7554/eLife.34465.003

only certain regions of olfactory epithelium were SHH+ (see stained cryosections under the 3D models in *Figure 2*). Later on (at E13.5 and E14.5 – *Figure 2—figure supplement 1*), we detected additional SHH-producing structures such as whiskers (blue-green), tongue (not segmented) and salivary

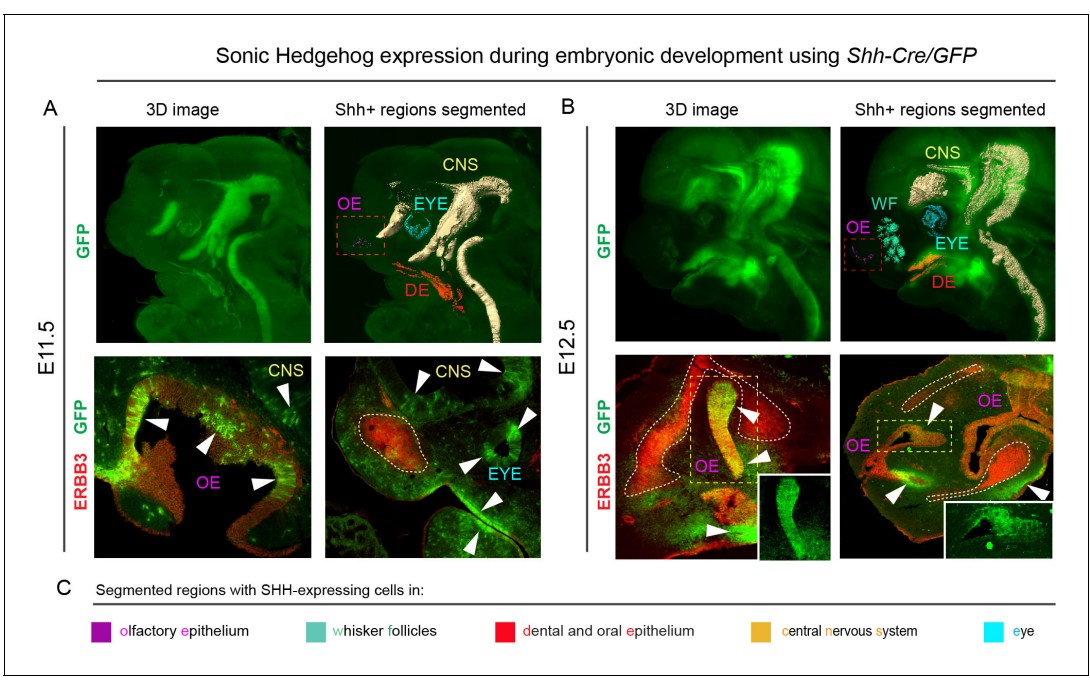

**Figure 2.** *Shh* is expressed through the early facial development in distinct regions of the head. The expression pattern of *Shh* during developmental stages E11.5 (A) and E12.5 (B) in *B6.Cg-Shhtm1(EGFP/cre)Cjt/J*, segmented *Shh*-expression regions are color-coded and clarified in legend (C). Immunohistochemical (IHC) staining shows olfactory neuroepithelium and newly formed mesenchymal condensation as ERBB3 positive. White arrows point at GFP-expressing parts of various tissues. Red rectangle in (A) and (B) upper panel mark the olfactory neuroepithelium, yellow rectangle in (B) shows area magnified in the right bottom corner. White dotted line outlines the shape of mesenchymal condensation. IHC staining was performed on 20 µm cryosections and imaged using a confocal microscope.
DOI: https://doi.org/10.7554/eLife.34465.005

The following figure supplement is available for figure 2:

**Figure supplement 1.** *Shh* is expressed through the later facial development in distinct regions of the head.
DOI: https://doi.org/10.7554/eLife.34465.006

gland (blue). We assumed that there is a prerequisite for a certain minimal distance between *Shh*-expressing structures and forming cartilage of nasal capsule that enables the secreted molecule to reach the target and impose chondrogenic stimuli on mesenchymal cells. According to our results, the most probable pro-chondrogenic SHH-emitting structures in the face were the olfactory epithelium and the forebrain.

To test if SHH emitted by these structures controls nasal capsule induction or influences its geometrical features, we analyzed a series of mouse mutants with a micro-CT-based 3D-visualization approach. To perform these 3D visualizations of the mesenchymal chondrogenic condensations, cartilage and bone, we utilized a soft tissue contrasting with phosphotungstic acid (PTA) followed by micro-CT scanning. Chelation of tungsten is uneven in various cell types and creates reliable contrast highlighting different tissues.

Firstly, to address whether the induction of distinct elements of the facial chondrocranium is not only timely regulated, but also has a discrete spatial aspect related to various sources of inductive signals, we genetically ablated *Shh* in the brain to test its role in facial cartilage induction. For this, we took advantage of *Nkx2.2-Cre/Shh* $^{floxed/floxed}$ animals to delete *Shh* in the floor plate cells since the beginning of central nervous system (CNS) development and patterning. We analyzed two different stages: E12.5 as a stage of condensations of cartilaginous mesenchyme (*Figure 3* and *Figure 3—figure supplement 1* 'Interactive PDF') and E15.5 as a later stage of nearly fully formed chondrocranium with its rather final shape (*Figure 3* and *Figure 3—figure supplement 1* 'Interactive PDF'). This experiment resulted in an unexpected phenotype. It included a selective loss of a nasal septum together with heavily affected posterior part of the nasal capsule as detected at E12.5 (*Figure 3B–C, E–F*). The chondrogenic condensation corresponding to a forming septum failed completely, whereas the condensation of the posterior part of nasal capsule appeared incompletely induced. Additional changes were detected in the facial cartilage at E15.5, mainly represented by the missing midline groove (*Figure 3G–H*) and the absence of the very anterior part of nasal cartilage (*Figure 3H*). Additional changes included various malpositioned parts, fused nerve foramina and variations in shaping and bending of cartilage elements, as summarized in *Figure 3*. Notably, the mutant embryos analyzed at E12.5 and E15.5 presented with cleft palate. This could indicate that some clinical cleft palate cases might have their origin in disturbed brain-derived signaling (*Figure 3—figure supplement 2*).

At the same time, the general geometry of the frontal part of facial chondrocranium remained almost unperturbed, thus, supporting the spatial modularity of cartilage induction in the face. The microstructure of the cartilage stayed normal, with fine borders defining bent cartilaginous sheets forming the major structure of the nasal capsule. The thickness of the cartilaginous sheets forming the capsule also remained comparable to that found in littermate controls (*Figure 3J*). These observations strongly suggested that the early stages of cartilage induction must be affected.

We also investigated bone formation in the area of the nasal capsule at E15.5 (*Figure 3I*). We observed missing parts in maxillary bones from mutants, and malpositioned incisors that were found more posteriorly on top of instead of being in the anterior part of the maxilla. At this stage and in this particular location, there was no endochondral ossification ongoing. However, according to micro-CT data, the bone was forming in the proximity and on top of the existing cartilaginous shape template. Thus, the facial chondrocranium is important for the correct formation of the membranous facial bones.

Recently it has been shown that the brain can influence facial shaping via *Shh*, acting presumably on the frontonasal ectodermal signaling zone (abbreviated as FEZ) (*Hu et al., 2015a*; *Chong et al., 2012*). It is currently believed that FGF8, SHH and BMPs produced by FEZ regulate the behavior of ectomesenchymal tissue and participate in positioning of chondrogenic condensations inside of the embryonic face (*Foppiano et al., 2007*; *Hu et al., 2015b*; *Young et al., 2014*). However, our data show that SHH emitted from the forebrain mostly affects the basicranial, posterior and septal parts of the facial chondrocranium without strong effects on the most anterior nasal capsule (including other soft tissues in general), which could be expected if the effects of a brain-derived SHH are entirely mediated by FEZ. Importantly, the major geometric structure of the mutant brain stays largely unchanged although it appears reduced in size. Thus, changes in brain shape are not likely to cause very selective influences on forming facial structures due to mere mechanical interactions (*Figure 3—figure supplement 3*).

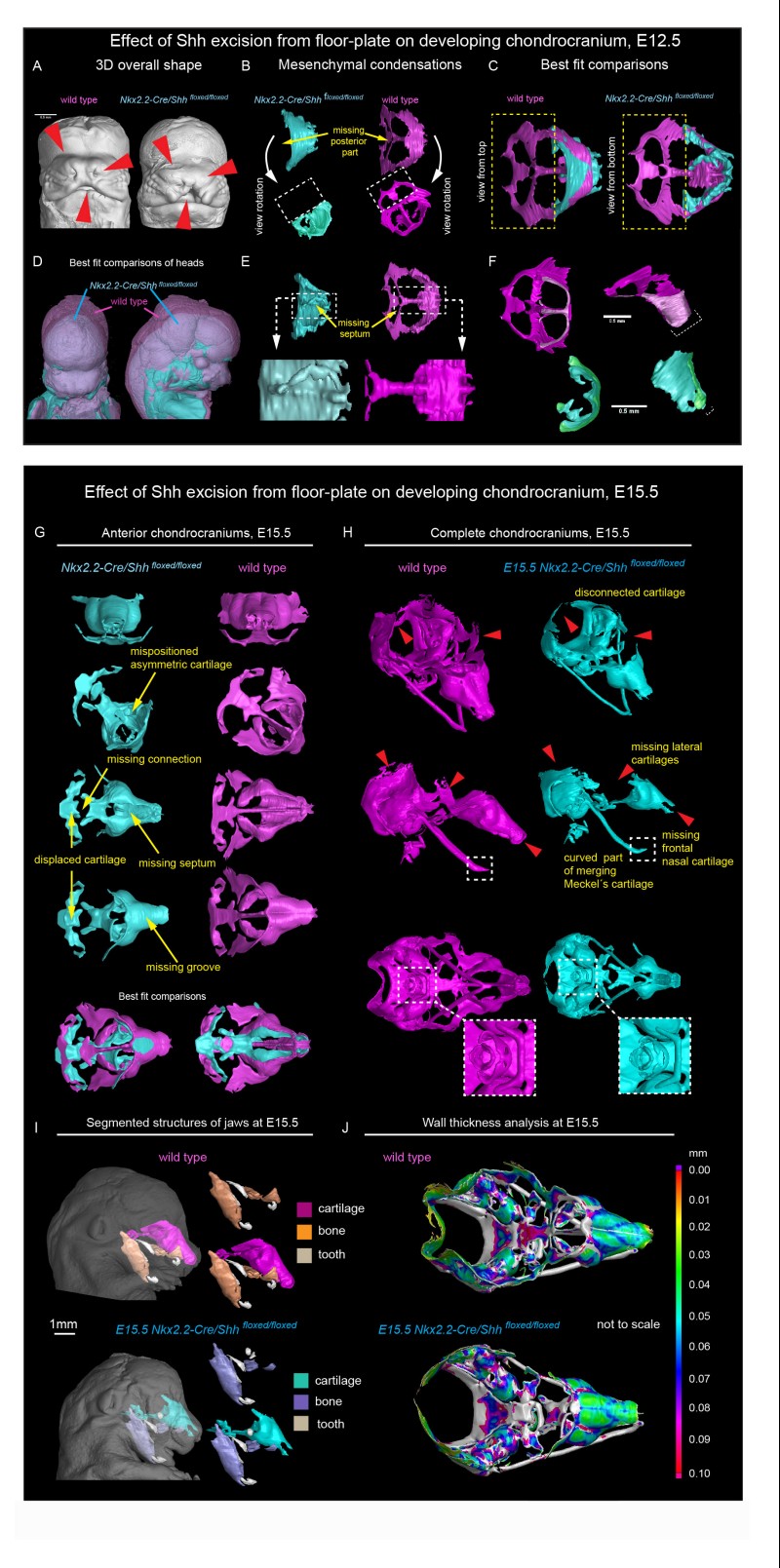

**Figure 3.** *Shh* signal from the brain induces a posterior part of a nasal capsule. (**A**) Model of overall shape resulting from the excision of *Shh* from the floor plate using *Nkx2.2-Cre/Shh*<sup>floxed/floxed</sup> shows visible phenotype in the E12.5 mutant embryo comprising, for instance, from the cleft of upper lip, non-prominent or missing nasal vestibule and diminished curvature of the snout. (**B–C, E–F**) micro-CT scans-based 3D reconstructions of chondrogenic mesenchymal condensations in E12.5 mutant and control embryos. Note the missing posterior part of the developing nasal capsule and the missing

*Figure 3 continued on next page*

*Figure 3 continued*

septum in the mutant. (**D**) best-fit computed comparison of the overall shape and size of mutant and control embryo. (**G–H**) 3D models of chondrocrania of mutant (*Nkx2.2-Cre/Shh^{floxed/floxed}*) and control embryo, analyzed at E15.5. Among the main differences are missing frontal part of nasal cartilage, missing lateral parts of developing nasal capsule, malpositioned asymmetric cartilage, not fully closed foramina for pervading nerves and vasculature and various disconnected cartilaginous segments. (**I**) Segmented cartilage and bones projected in the overall shape of the head of mutant (bottom) and control (upper) embryo. Note the malpositioned incisors in the maxilla of the mutant and missing part of the frontal nasal capsule formed by the bone. (**J**) Wall thickness analysis of the cartilages in the E15.5 head of mutant (bottom) and control (upper) embryo show no evident differences in the thickness of formed cartilage.

DOI: https://doi.org/10.7554/eLife.34465.007

The following figure supplements are available for figure 3:

**Figure supplement 1.** 'Interactive PDF'.
DOI: https://doi.org/10.7554/eLife.34465.008
**Figure supplement 2.** Ablation of *Shh* from the floor plate results in the cleft palate.
DOI: https://doi.org/10.7554/eLife.34465.009
**Figure supplement 3.** Brain volume and overall anatomy in mutants with *Shh* genetically deleted from the floor plate of the developing CNS.
DOI: https://doi.org/10.7554/eLife.34465.010
**Figure supplement 4.** Phenotypic manifestation of *Shh* genetic ablation from the floor-plate, analyzed at E15.5 upper part of spine.
DOI: https://doi.org/10.7554/eLife.34465.011

Finally, we analyzed the spinal column of the mutants and control embryos and found localized shape defects in the cervical vertebrae represented by incomplete transverse foramina (*foramina transversariae*) (*Figure 3—figure supplement 4*). Notably, the defects in the nasal septum stayed confined only to the cartilage as a tissue. The septal chondrogenic condensations and cartilage were missing as apparent E12.5 and E15.5, whereas the soft tissues of the septum stayed in place (*Figure 3—figure supplement 4*). Similarly, despite the cartilage defects clearly identified in spinal column, the other tissue types that were in close proximity to the defects did not show any difference from control. For instance, the vertebral arteries traversing the distorted cartilage of the transverse foramina appeared unaffected in *Nkx2.2-Cre/Shh ^{floxed/floxed}* animals (*Figure 3—figure supplement 4*).

Next, we determined the *Shh* expression in the forming olfactory epithelium and tested the role of the forming olfactory epithelium and olfactory neurons in the process of nasal capsule induction. The possibility that olfactory epithelium controls cartilage shaping is supported by the fact that the conchae of nasal labyrinth geometrically correlate with the folded structure of the olfactory epithelium, which they mechanically support.

A desired experimental setup that would allow us to target and genetically delete *Shh* from the olfactory epithelium does not exist. To define the importance of any signal derived from the olfactory epithelium we utilized *Six1* and *Six4* double knockout to specifically ablate the development of the olfactory epithelium (*Ikeda et al., 2007*; *Kobayashi et al., 2007*). We analyzed *Six1/4* double mutant embryos at E18.5, that showed no olfactory structures, using the micro-CT method and performed a thorough comparison to wildtype littermates (*Figure 4* and *Figure 3—figure supplement 1* 'Interactive PDF'). In these mutants which had markedly shortened noses, the roof of the nasal capsule was entirely missing. However, the septum and the posterior part of the capsule were relatively well-preserved (*Figure 4B–D*). An earlier analysis at E12.5, a stage when the majority of facial chondrogenic condensations come into place, showed that in double knockouts, the chondrogenic condensation corresponding to the nasal capsule roof is missing. The chondrogenic condensation corresponding to the nasal septum appeared non-fused at this stage and fused only later, as evident from E17.5 reconstructions. At the same time, the roof of the nasal capsule was never induced (*Figure 4—figure supplement 1A–D*). This is consistent with the prediction that a lack of the olfactory epithelium will prevent the induction of olfactory cartilages. Analyses of the bones showed a major change in the overall geometry, with prominent shortening along the anterior-posterior axis (brachycephalic-like appearance). Furthermore, we noticed a lack of mandibular incisors and an obvious asymmetry between left and right parts of the maxilla (*Figure 4E,J*).

In addition to the expression in olfactory placodes, *Six1* and *Six4* are expressed in different parts of early facial mesenchyme (*Kobayashi et al., 2007*; *Grifone et al., 2005*). The full knockout of *Six4* does not show any phenotype according to previously published data (*Ozaki et al., 2001*). We

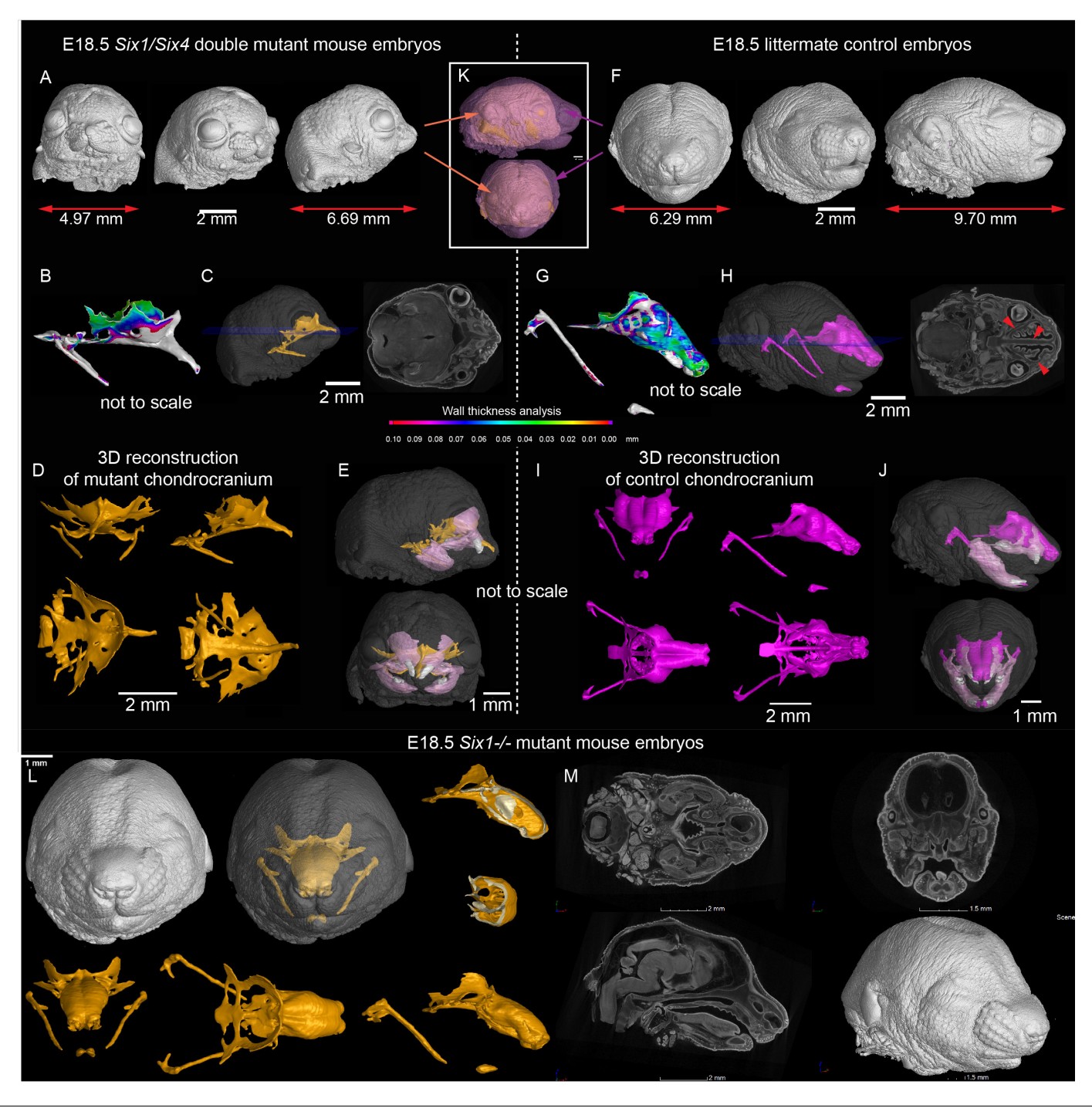

**Figure 4.** Signals from the olfactory epithelium induce the anterior part of the nasal capsule. (A–J) *Six1/Six4* double knock out mouse embryo compared to littermate control and analyzed at E18.5 using micro-CT and 3D-reconstruction. (A, E, K) overall shape and size of the mutant head is different from the control head, showing brachycephalic phenotype, bulging eyes, misshaped earlobes positioned more frontal and lower than the control embryo (F, J, K). Note also the left-right asymmetry of the snout of the mutant. (B, G) Wall thickness analysis of mutant (B) and control (G) embryo. (C, H) single plane from raw CT data shows missing olfactory neuroepithelium in the *Six1/Six4* double mutant and control. Note also missing nasal capsule but retained septum in the mutant. (D–E, I–J) various views on segmented 3D model of chondrocranium of mutant (D–E) and control (I–J) embryo. Among the obvious differences are missing roof and lateral parts of nasal capsule while the septum is preserved. (L) *Six1* single knock out mouse embryo analyzed at E18.5 using micro-CT 3D reconstruction (M) single plane cross-sections from raw CT of *Six1* single knock out E18.5 embryo.
DOI: https://doi.org/10.7554/eLife.34465.012

*Figure 4 continued on next page*

*Figure 4 continued*

The following figure supplements are available for figure 4:

**Figure supplement 1.** Analysis of mesenchymal condensations in *Six1/Six4* double knock out embryos at E12.5.

DOI: https://doi.org/10.7554/eLife.34465.013

**Figure supplement 2.** *Ascl1* knockout embryo analyzed at E16.5 does not show any significant changes in formed nasal capsule.

DOI: https://doi.org/10.7554/eLife.34465.014

therefore analyzed the full knockout of *Six1*, in which the olfactory epithelium is present, to see whether it will show a mesenchyme-driven phenotype in the nasal capsule roof. The analysis of E18.5 *Six1$^{-/-}$* embryos demonstrated the presence of the nasal capsule roof as well as a septum. The phenotype in *Six1$^{-/-}$* embryos mostly included a narrowing of the posterior nasal capsule with mild septal defects (*Figure 4L–M* and *Figure 4—figure supplement 1E*).

This experiment provided a complementary, non-overlapping phenotype to the one with a nasal capsule resulting from the excision of *Shh* from the floor plate (*Nkx2.2-Cre/Shh$^{floxed/floxed}$*). This suggests that solid cartilage elements in the forming face depend on joint activities of multiple regulatory zones (sources of SHH) during their induction and shaping.

To investigate if sensory neurons in the olfactory neuro-epithelium might be potential sources of SHH, we utilized *Ascl1* (*Mash1*) knock-out embryos (*Figure 4—figure supplement 2*). In these animals, major neurogenic transcriptional factor essential for olfactory neuron formation are missing, and very few olfactory sensory neurons are generated (*Guillemot et al., 1993*; *Cau et al., 1997*). We analyzed craniofacial structures in these mutants and found out that they did not demonstrate any gross abnormality despite their inability to develop large amounts of olfactory neurons. Hence, the sensory olfactory neurons are not critical cartilage-inducing sources, while the early olfactory epithelium, before the neurogenesis, is important for the formation of chondrogenic mesenchymal condensations.

To check if SHH from the brain and presumably from the olfactory epithelium acts directly on facial mesenchyme inducing chondrogenic differentiation or during cartilage growth, we analyzed embryos carrying a SHH-activity reporter *GBS-GFP* (*Balaskas et al., 2012*) at different developmental stages ranging from E9 to E14.5. The GFP signal was detectable as expected in the forming palate, brain, spinal cord and tissues that are known to receive SHH input. However, we did not observe GBS-GFP activity in the chondrogenic mesenchymal condensations from the earliest chondrogenic stages E11.5-E12.5 onwards (*Figure 5A*). Consistently with this, lineage tracing in *Gli1-CreERT2/R26Tomato* mice injected with tamoxifen at E11.5 and analyzed at E17.5 showed sporadic patches of traced chondrocytes in the facial cartilages (*Figure 5B*). If the injection of tamoxifen was performed at E12.5 or later, these sporadic patches of labeled chondrocytes disappeared; that is we observed only very rarely scattered chondrocytes in other locations (*Figure 5B–C*). Analysis of the rare individual clones of chondrocytes resulting from labeling at E11.5 in *Gli1-CreERT2/R26Confetti* demonstrated that mesenchymal cells turn into chondroprogenitors that divide several times to generate clonal clusters of mature chondrocytes (*Figure 5C–D*). At later stages of tamoxifen administration, this was not observed, which is consistent with analysis of *Gli1-LacZ* embryos at E12.5, where the X-gal staining was confined to whisker pads and other peripheral locations. In situ hybridization (RNAscope) for other components of the Hedgehog pathway (including *Ptch1*(mRNA), *Ptch2* (mRNA), *Gli1*(mRNA), *Gli2*(mRNA)) showed no enrichment within potential and actual chondrogenic areas (*Figure 5—figure supplement 1*). To functionally test if inhibition of SHH at chondrogenic stages will affect the development of facial cartilage, we administered SHH-inhibitor vismodegib (*LoRusso et al., 2011*) at either E11.5, E12.5, or E13.5 and analyzed the embryos at E15.5. In line with predictions from expression analysis and SHH-activity reporter, we did not find striking abnormalities in nasal and Meckel´s cartilages from all treated and analyzed embryos (n = 4) (*Figure 6A*). At the same time, we observed the absence of palate and concomitant abnormalities in whiskers distributions in E11.5-to-E15.5 and E12.5-to-E15.5 stages of treatment, but not in E13.5-to-E15.5 stages. Indeed, in embryos treated between E11.5 and E12.5, the palatal shelves were severely reduced or missing (*Figure 6A–C*). This showed that SHH signaling in these embryos was inhibited to significant extents and also suggested that the action of SHH on patterning of the nasal capsule precedes the stage of chondrogenic condensations.

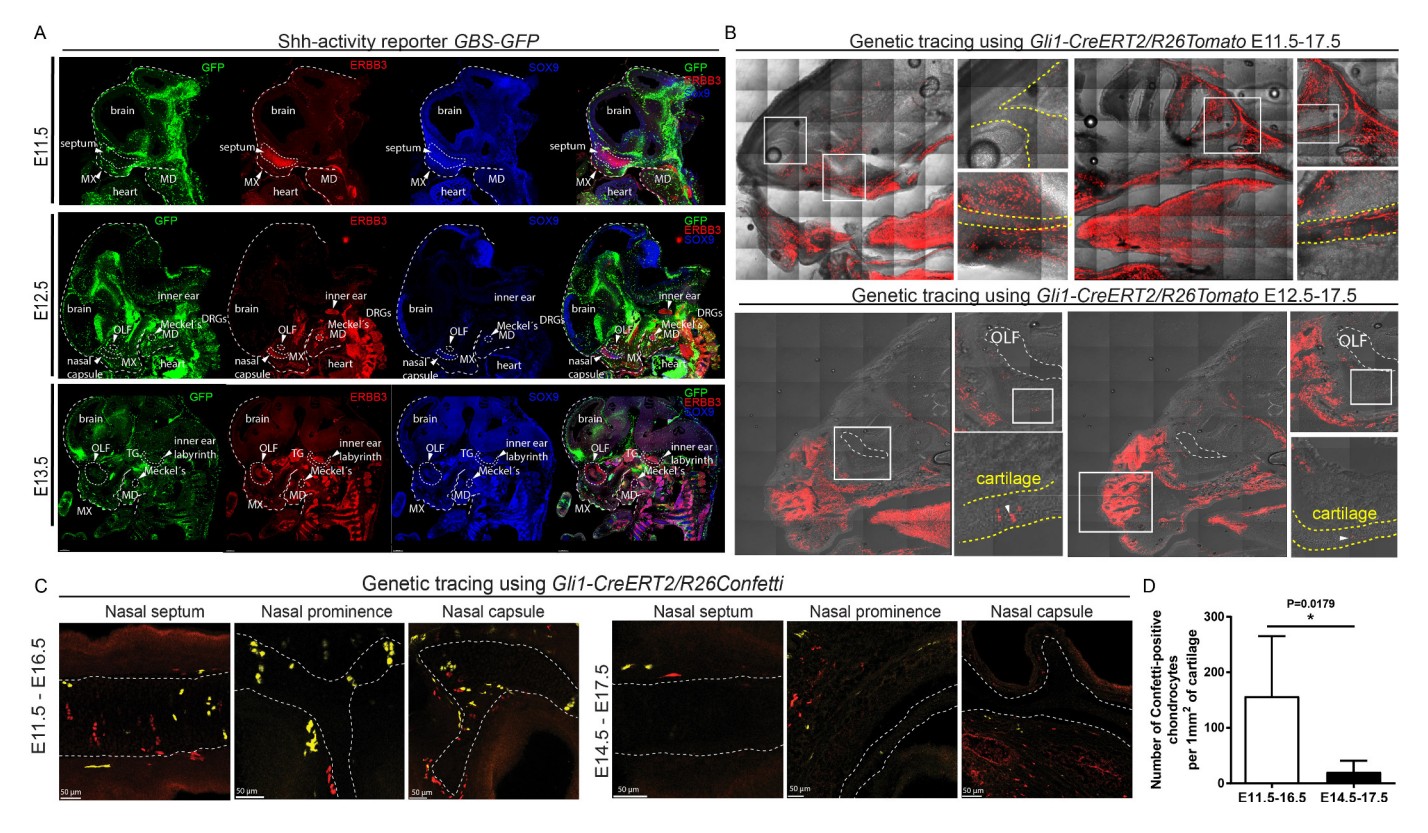

**Figure 5.** Mapping of the expression of Gli1 and Shh signaling activity in mouse embryonic head. (**A**) Mapping of the activity of the SHH signaling in mouse embryonic head at E11.5, E12.5 and E13.5 prechondrogenic and early chondrogenic stages using *GBS-GFP* activity reporter. (**B**) Genetic tracing using *Gli1-CreERT2/R26Tomato* induced at E11.5 (upper panel) and E12.5 (bottom panel) and analyzed at E17.5. Dotted line outlines cartilaginous structures within the nasal capsule. White squares outline the magnified area. DRG = dorsal root ganglion, OLF = Olfactory system, MX = maxillar prominence, MD = mandibular prominence. TG = trigeminal ganglion. (**C**) Genetic tracing using *Gli1-CreERT2/R26Confetti*, induced at E11.5 and analyzed at E16.5 (left panel) and induced at E14.5 and analyzed at E17.5 (right panel) shows the contribution of Gli1-traced positive cells to the cartilaginous structures in the embryonic head. (**D**) Quantification of the contribution of Gli1-traced positive cells to the cartilage. (**A–C**) 20 μm cryosections were used for the IHC staining and analysis. A confocal microscope has been used for imaging.

DOI: https://doi.org/10.7554/eLife.34465.015

The following figure supplement is available for figure 5:

**Figure supplement 1.** Mapping of the presence of major SHH signaling components in the E12.5 embryo.
DOI: https://doi.org/10.7554/eLife.34465.016

Consistent with these results, which suggest an early pre-chondrogenic role of SHH signaling on nasal capsule patterning, the analysis of embryos homozygous for a hypomorphic *Shh* allele (*Shh-GFP*, here referred to as *Shh*[Hypo]), in which SHH signaling is constitutively reduced (*Zagorski et al., 2017*; *Chamberlain et al., 2008*), revealed severe abnormalities in the facial cartilage (*Figure 6D* and *Figure 6—figure supplement 3*). The results of all genetic perturbations and treatments with drugs as well as their comparative phenotypes are summarized in *Figure 7*. Taken together, the effects of SHH on chondrogenic differentiation in the facial region are early and precede the first wave of chondrocyte differentiation that occurs between E11.5 and E14.5. These results are also consistent with the phenotype of *Six1/Six4* double knockout embryos at E12.5 (*Figure 4*), and corroborate the notion of an early pre-cartilage onset of the phenotype.

Tissue-specific expression of *Shh* is known to be controlled by multiple enhancers. Some, which may regulate *Shh* expression in the cranial region, have been characterized (*Yao et al., 2016*; *Jeong et al., 2006*; *Sagai et al., 2009*). To modulate *Shh* expression in different craniofacial regions, we analyzed mutant mice carrying different rearrangements (deletions or inversions) with impacts on the distal regulatory landscape of *Shh* (*Symmons et al., 2016*). We paid specific attention to

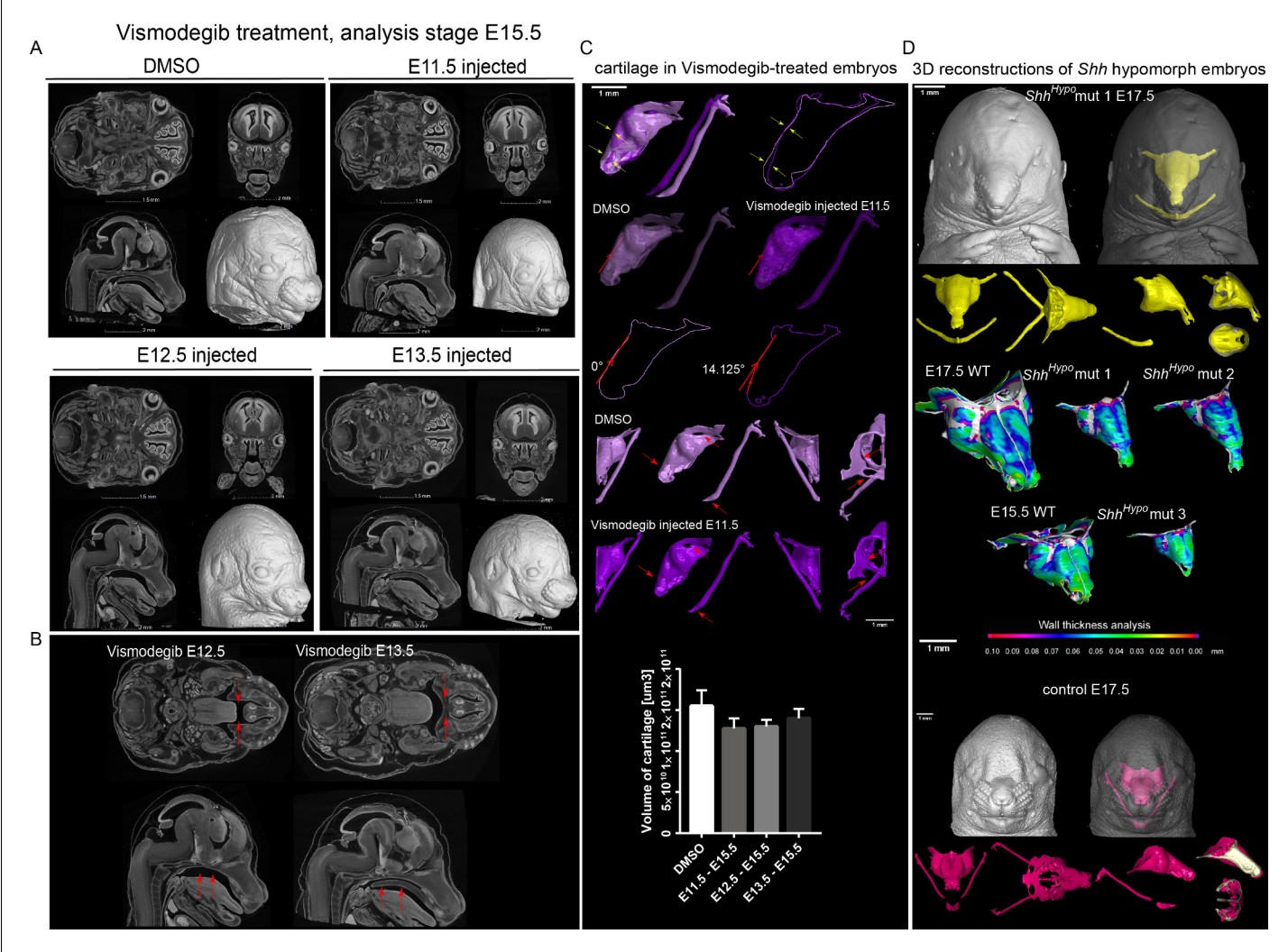

**Figure 6.** Effect of Vismodegib treatment on the size and structure of nasal cartilage. (**A**) Panel shows raw CT data, cross-sections from various planes from DMSO (control treatment), Vismodegib inhibitor administered at E11.5, E12.5 or E13.5, all analyzed at E15.5 (**B**) Raw CT cross-sections show absent/disrupted cartilaginous structures in Vismodegib-treated embryos (**C**) 3D reconstruction and comparison of inhibitor-treated (Vismodegib E11.5-E15.5) and control (DMSO-treated) embryos and their nasal capsules. Arrows point at differences in anatomy and overall geometry of the nasal capsule. Graph representation of overall cartilage change of inhibitor-treated embryos. Raw data are available in *Figure 6—source data 1*. (**D**) 3D-reconstruction of the nasal capsule of *Shh* hypomorph (analyzed at E15.5 and E17.5) including wall thickness analysis.

DOI: https://doi.org/10.7554/eLife.34465.017

The following source data and figure supplements are available for figure 6:

**Source data 1.** Raw values of cartilage measurements corresponding to Graph in *Figure 6C*.
DOI: https://doi.org/10.7554/eLife.34465.021

**Figure supplement 1.** Mutations of various regulatory regions controlling expression of *Shh*, their positions and effect on chondrogenesis.
DOI: https://doi.org/10.7554/eLife.34465.018

**Figure supplement 2.** Mutated genomic regions containing regulatory sequences controlling expression of *Shh* show a variety of similar and dissimilar phenotypes.
DOI: https://doi.org/10.7554/eLife.34465.019

**Figure supplement 3.** The effect of reduced SHH signaling on chondrogenesis at E15.5 and E17.5.
DOI: https://doi.org/10.7554/eLife.34465.020

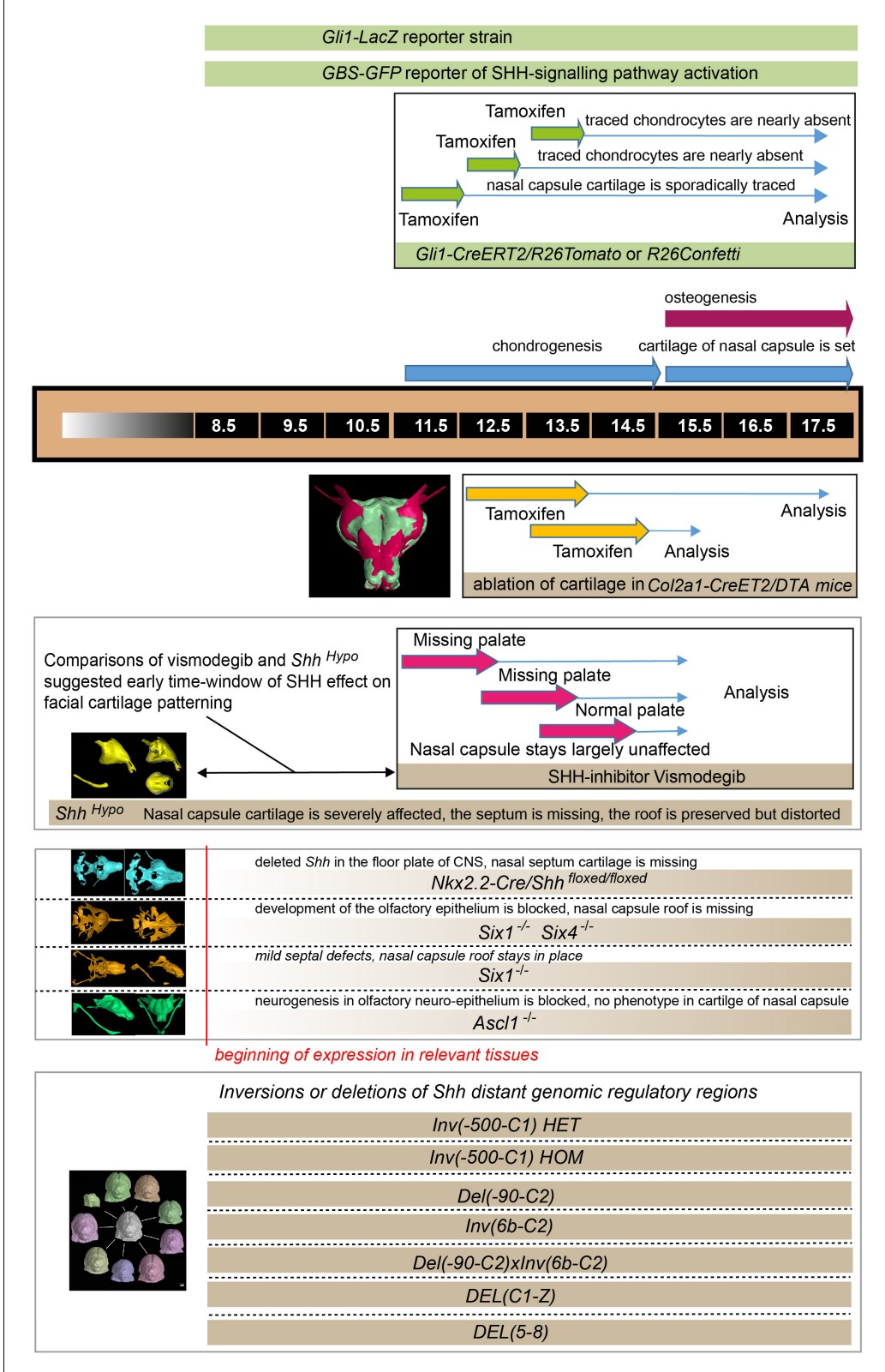

**Figure 7.** lustration depicting the timeline of all experiments and the beginning of effects. The results of all genetic perturbations and treatments with drugs as well as their comparative phenotypes are summarized.

DOI: https://doi.org/10.7554/eLife.34465.022

genomic regions where known and potentially unknown brain-specific enhancers (SBE, Shh Brain Enhancers) are located (*Jeong et al., 2006*; *Sagai et al., 2009*). Currently known SBEs (SBE2, SBE3, SBE4) localize between the *Shh* coding region and *mark 6b* (*Irimia et al., 2012*) (see scheme in *Figure 6—figure supplement 1*). Animals carrying an allele with the inversion of a large genomic region including known SBEs, *INV(−500 C1)*, demonstrated selectively localized mild defects in the nasal septum and also in the tips of Meckel's cartilage, but not in the roof of nasal capsule. This septum-specific phenotype appeared more pronounced in a homozygous *INV(−500 C1)* as compared to the heterozygous genotype. Interestingly, this allele does not change the relative position to *Shh* of most known enhancers that could be responsible for SHH-dependent facial skeletal development. The phenotype observed in this experiment showed that septal cartilage is sensitive to minor changes in SHH patterning signals that result from either removal of regulatory sequences distal to the C1 position, or indirect effects affecting distant enhancer-promoter- communications.

*INV(6b-C2)*, which removes additional enhancers required for ventral forebrain expression of *Shh* in E10.5 embryos (*Figure 6—figure supplement 1A–C*), showed much stronger phenotypes.

Such strong phenotypes are most likely due to disruption of the *Shh* TAD (Topologically Associating Domain), and displacement of major regulatory elements at a position which prevent their interaction with *Shh* (*Symmons et al., 2016*). Accordingly, *Shh* expression is much reduced in the forebrain of mutant embryos at E10.5 as evident from the results of in situ hybridization for *Shh* (*Figure 6—figure supplement 1A–C*).

Heterozygote *INV(6b-C2)* did cause minor defects in septum, while the same inversion over a full Shh deficiency (*DEL(−90 C2)*) led to a strong holoprosencephaly similar to *Shh^Hypo* (*Figure 6* and *Figure 6—figure supplement 2*. However, it was not as strong as a full *Shh* knockout, where only a proboscis is left.

In the mutants *INV(6b-C2) x DEL(−90 C2)*, the nasal septum did not develop at all, and the rest of the nasal capsule (roof) was present although severely disturbed in shape. The gradual increase in severity of the malformations from *INV(−500 C1)* to *INV(6b-C2)* may indicate the presence of several distinct enhancers related to face morphology distal to position 6b (which likely includes yet uncharacterized enhancers). It could also mean that the efficiency of the interactions of the known enhancers (SBE2-4) with the *Shh* promoter is modulated by the presence of distal elements, which may contribute to organize the *Shh* TAD (*Symmons et al., 2016*). Both scenarios suggest the existence of other regulatory regions important for facial development including new potential SBEs based on the reduction of the *LacZ*-based *Shh*-expression reporter signal in the forebrain of mutant embryos at E10.5 (*Figure 6—figure supplement 1*).

*DEL(5-8)* showed a powerful phenotype similar to the homozygous *INV(6b-C2)*. This, in a combination with previous phenotypes, reveals that the key regulatory regions essential for the facial cartilage patterning are located between *marks 6b* and *C1*. These regulatory regions are likely SBEs since the LacZ expression disappears from the forebrain of reporter embryos at E12.5 according to previous results (*Symmons et al., 2016*). *DEL(C1-Z)* did not show any gross abnormalities in the nasal capsule cartilage. Embryos carrying this mutation showed abnormal Meckel's cartilage and a generally affected mandible.

Taken together, local tuning of SHH expression by various enhancers (including brain-specific regulatory elements) seemingly controls discrete steps of chondrocranial patterning, which may represent a key evolutionary mechanism shaping animal snouts and faces.

## Discussion

In addition to the obvious functional aspects, facial shape is important in many ways. For example, recognition of individual facial features assists social interaction and affects numerous important aspects of our everyday life (*Vernon et al., 2014*). Pathological conditions include a very wide spectrum of deficiencies, and may involve eating, breathing and speech impairments, emotional problems and low quality of life in general (*Forbes, 2010*). Here, we demonstrated that even fine and selective manipulations of a facial cartilage geometry and size (performed in *Col2-CreERT2^{+/-}/ R26DTA^+* embryos) result in significant changes of adjacent membranous bones and facial shape. In turn, the facial cartilage geometry is controlled by the signals coming from neurosensory structures such as brain and olfactory epithelium. Altogether, these results might provide a new explanatory

framework revealing how the integrated development of neural and non-neural tissues results in the precise and evolutionary adapted shape of the bony cranium and corresponding facial appearance.

Previously, Marcucio and Hallgrimsson laboratories demonstrated the co-variation of brain and face as well as the impact of brain-emitted SHH on avian facial development (*Hu and Marcucio, 2009*; *Marcucio et al., 2011*; *Parsons et al., 2011*; *Petryk et al., 2015*). Yet, the role of such brain-derived signals in cartilage and bone shaping has not been extensively investigated. According to our results, SHH, a key signal enabling cartilage induction, arrives from the floorplate of the brain, and the selective ablation of *Shh* in that area by non-inducible *Nkx2.2-Cre* causes a highly selective loss of the nasal septum cartilage. The soft non-cartilaginous tissue of the nasal septum, however, remains intact, rendering this part of the phenotype highly cartilage-specific. The roof of the nasal capsule stays nearly unaffected. The analysis at E12.5 stage suggested that this phenotype must be related to pre-chondrogenic, early SHH-dependent patterning signals enabling cartilage formation.

This reasoning is further supported by the line of experiments involving 'early on action' $Shh^{Hypo}$ embryos, which demonstrated a profound phenotype in the nasal capsule cartilage. This was in contrast to wild type embryos treated with the SHH-inhibitor Vismodegib strictly between chondrogenic stages (E11.5-E13.5). These embryos showed no significant phenotype in the nasal capsule cartilage. Correspondingly, lineage tracing with *Gli1-CreERT2* and expression analysis of SHH pathways members starting from E11.5 did not show any association with development of facial cartilages.

These cartilage-related results, where SHH-activity was not associated with facial chondrogenic differentiation, were dramatically different from those observed in palate-forming mesenchymal cells in the same experimental embryos. Unlike cartilage, the developing palate showed strong activity of the SHH-signaling reporter *GBS-GFP*. We also found high expression levels of SHH pathway members, including numerous traced progeny in *Gli1-CreERT2/R26Tomato* animals. As expected, in embryos treated with Vismodegib between E11.5-E12.5, the palate was almost missing, in contrast to contrary to the nasal capsule cartilage that stayed virtually intact. The presence of abnormal palate clefts in Vismodegib-treated and also in *Nkx2.2-Cre/Shh* $^{floxed/floxed}$ embryos agrees with previous findings, which have established a general role of SHH in the patterning of the face and in development of pathological midfacial clefts (*Hu and Helms, 1999*). Accordingly, ciliopathies and their animal models often demonstrate similar defects (*Brugmann et al., 2010*; *Chang et al., 2016*).

Thus, according to our results, SHH is not involved into facial chondrogenesis at stages when chondrogenic condensations form and convert into mature cartilage. Hence, the role of SHH is most likely indirect. Presumably, it is involved in the early patterning of facial ectomesenchyme, to create proper conditions for the sophisticated facial chondrogenesis that will start at later developmental stages. In line with this reasoning, the mode of action of SHH on chondrogenesis in non-facial locations, for example, during the development of axial skeleton, is indirect and based on SHH-dependent alterations of cell responses to BMPs in potentially pre-chondrogenic mesenchymal cells. Murtaugh et al. demonstrated that even a transient SHH signal was able to ensure the competent chondrogenic response of mesenchymal cells to BMPs (*Murtaugh et al., 1999*). In the case of the axial skeleton, such competence-changing action of SHH depends on the initiated and sustained expression of the transcriptional repressor *Nkx3.2*, which renders cells responsive to pro-chondrogenic actions of BMPs (*Murtaugh et al., 2001*). As shown by Zeng et al., expression of *Nkx3.2* is sufficient to induce SOX9, a major chondrogenic master regulator, and in the presence of BMPs, NKX3.2 and SOX9 could induce the expression of each other (*Zeng et al., 2002*). Finally, the sequential action of SHH and BMPs could induce SOX9. According to the public in situ database Allen Developmental Mouse Brain Atlas (http://developingmouse.brain-map.org), *Nkx3.2* shows distinct expression in a range of cranial cartilages or their future locations at pre-chondrogenic (E11.5) and chondrogenic (E13.5) stages. The knockout of *Nkx3.1* and *Nkx3.2* yields changes in the facial shape, as evident from *Figure 1* in Herbrand et al. (*Herbrand et al., 2002*). Similar patterning effects of SHH in relation to chondrogenesis have previously been demonstrated during the development of serial tracheal rings reinforced with cartilage. SHH inactivation in ventral trachea resulted in a lack of tracheal segmentation which coincided with the loss of cartilage, while the upregulation of *Shh* resulted in cartilage overproduction and similar segmentation defects (*Sala et al., 2011*). Altogether, this may suggest an analogous or similar mechanism of an indirect action of SHH on craniofacial chondrogenesis, especially in light of our results showing only minor and sporadic activity of SHH-signaling reporter in facial chondrogenic condensations or cartilage. Determination of the mechanisms behind SHH action in facial chondrogenesis, with a special focus on the position of the SHH-

signal in the hierarchy of events leading to chondrogenic differentiation, is a key future direction. We anticipate that Single cell transcriptomics approaches (*Picelli et al., 2013*) will be applied to facial mesenchymal populations prior and during chondrogenesis. This should resolve cell signaling cascades with high precision along the developmental timeline, in similarity to pioneering studies utilizing this methodology in other tissue types (*Furlan et al., 2017*).

Complementary to the ablation of *Shh* by *Nkx2.2-Cre*, the loss of the olfactory epithelium in *Six1/Six4* double knockout mutants leads to the absence of the nasal capsule roof, while the nasal septum cartilage stays largely in place. Our data indicate that the loss of a nasal capsule roof in the double knockout condition is driven by the loss of nasal placodes (requiring both SIX1 and SIX4), which causes a collapse of olfactory epithelium. Despite the specific loss of nasal capsule roof, other cartilages appeared intact in locations corresponding to the expression sites of *Six1* and *Six4* (including Meckel's cartilage). The perfect match between the shape of cartilaginous olfactory turbines and the epithelium, as well as the coordinated time course of their development (*Kaucka et al., 2017*), additionally support the notion that signals from the developing olfactory epithelium might enable the induction of adjacent cartilage. Still, these arguments cannot completely rule out the possibility that co-expressed *Six1* and *Six4* may have early roles during neural crest migration and early post-neural crest stages that might be important for the nasal roof formation. Taken together, our results provide strong support for the idea that a single solid cartilaginous element such as the nasal capsule can be induced by the combinatorial action of signals derived from several, in the present case, neural and neurosensory, locations. Notably, facial chondrogenic condensations are induced being 'pre-shaped'. Already at the earliest steps, they are laid down as highly complex 3D-geometries (*Kaucka et al., 2017*). The induction of such 3D-shapes is unlikely to be achieved by signals from a single site and might require more sources including spatially opposed brain and olfactory structures. Since we were unable to validate that SHH from the olfactory placode or epithelium is the key factor that induces a nasal capsule roof, we cannot exclude that other signaling molecules participate during critical steps of facial cartilage induction. This will require further investigations.

During evolution of vertebrates, cartilages forming the neurocranium and the future upper jaw appear before Meckel's cartilage attains a function of a lower jaw skeleton, and the animals acquire articulated hinged jaws (*Shimeld and Donoghue, 2012*). Therefore, one of the primeval functions of the neurocranial and frontal cartilages could be the encasement and protection of the neural and sensory compartments such as brain, eyes, ears and olfactory neurons. If that is the case, it is logical to reason that these neurosensory structures could emit cartilage-inducing signals and coordinate cartilage growth and shaping. Our experimental results reveal the key role of SHH from the developing brain in enabling the induction of a nasal capsule and basicranial cartilages, and, thus, support the aforementioned evolutionary hypothesis. The capacity of the developing olfactory epithelium to shape the cartilaginous support also favors this reasoning. Genetics-based prevention of neuronal differentiation in the olfactory epithelium (vial *Ascl1* knockout) does not interfere with shaping of the nasal capsule and confines the shaping role of presumably olfactory progenitors to the developmental period before their differentiation into the mature olfactory neurons. Alternatively, other cell types in the olfactory epithelium may play a cartilage-inducing role (olfactory glia, non-neurogenic epithelium).

In addition to the evolutionary aspect, the role of different neurosensory structures (mainly the brain) in coordinated cartilage induction may suggest a new connection between neurological and craniofacial symptoms in numerous genetic syndromes. Examples of such conditions are Williams syndrome, Down syndrome and others that are manifested by behavioral and morphological abnormalities in the central nervous system (*Starbuck et al., 2017*; *Weisman et al., 2017*; *Vincent et al., 2014*; *Antshel et al., 2008*) (and reviewed by [*Marcucio et al., 2011*]). Based on this reasoning, it is possible to envision a mechanistic connection between the fine aspects of a facial geometry and individual features of the human brain. An enormous facial variability is found among humans, which poses a question regarding the molecular and cellular mechanisms that underlie this variability. In difference to humans, non-human primates generally use variations in colored facial hairs to express their species, social status and sex in addition to body movements, voice expressions and scent (*Santana et al., 2012*; *Allen and Higham, 2015*). This brings us to speculate whether the loss of dense facial hairs during evolution of humans led to the development of a very broad range of various facial tissue-related features in order to compensate for the loss of facial hair-related communication and individual recognition. We hypothesize that one of those shape-tuning mechanisms could

include flexible and individual modulation of SHH, an important patterning and shaping agent during the embryonic development that comes from different spatial sources including the developing brain.

Previous research has established the existence and position of some of the specific regulatory elements that direct the expression of *Shh* in the craniofacial epithelial linings (*MRCS1* and *MFCS4*, see (*Sagai et al., 2009*) for details) as well as in the floor plate and anterior forebrain (*SBE2* ((Shh Brain Enhancer 2)), *SBE3* and *SBE4* [*Yao et al., 2016*; *Jeong et al., 2006*]). In order to find out fine effects in facial cartilages as a result of activity by site-specific enhancers, we analyzed a variety of mutants with deleted and inverted genomic regions containing such regulatory elements (mutants created by François Spitz' laboratory [*Symmons et al., 2016*]). The inversion of the −*500* C1 genomic region including known CBEs showed localized defects within the nasal septum that incrementally increased from heterozygous to homozygous state without influencing nasal capsule roof shape. Similarly, the inversion of (*6b-C2*) region in a heterozygous state caused minor defects in septum, whereas the same inversion on the background of deletion of the entire regulatory region (−*90 C2*) appeared similar to *Shh^Hypo* or even *Shh* knock out. This is explained by the translocation of TAD (see (*Symmons et al., 2016*) for details) and resulting 'isolation' of *Shh* coding part from head-specific regulatory regions located between positions six and *C2*. In this latter case (*INV(6b-C2)*), *Shh* expression was dramatically reduced in the anterior forebrain as compared to the control (*Symmons et al., 2016*).

Further analysis of more restricted *Shh* regulatory regions revealed that brain-specific and facial cartilage-related enhancers are confined within the region (*5-8*) and are at least partly responsible for the expression of *Shh* in the forebrain according to the loss of in situ hybridization signal in the forebrain of (*DEL(5-8)*) compared to controls. The deletion of this (*DEL(5-8)*) region resulted in severe facial malformation and collapse of the nasal capsule shape to the state resembling *Shh^Hypo*. Despite that we clearly observed the misshaped nasal capsule roof in these embryos, the septum was completely gone similarly to (*DEL(−90 C2) x INV (6b-C2)*) mutants. Taken together, these results provide strong support to the discrete role of genomic regulatory regions directing the expression of *Shh* to the forebrain and, through this, affecting the patterning of septal, basicranial and other cartilages in the head.

Importantly, all analyzed embryos carrying mutated regulatory regions never demonstrated missing nasal capsule roof including severe (*DEL(−90 C2) x INV (6b-C2)*), (*DEL (5-8) HOMO*) and *Shh^Hypo*. This might mean that we still do not know about the position of the corresponding regulatory regions targeting the expression of *Shh* to the olfactory epithelium or FEZ in the frontal face. The loss or inversion of regulatory regions resulting in mild septal defects did not affect the anterior nasal capsule, which might be independently patterned by FEZ. Similarly, the morphology of the anterior nasal capsule stayed relatively stable when septal cartilage disappeared in *Nkx2.2-Cre/Shh ^floxed/floxed* animals. This suggests that the most anterior face including frontal facial cartilages might be indeed patterned by FEZ together with olfactory placodes independently from brain-derived signals. These results point towards the possibility that mouse FEZ can form and act independently of the CNS signaling center contrary to chick embryonic development (*Hu and Marcucio, 2009*).

To summarize, it is possible that enhancer-dependent spatial and temporal regulations of *Shh* expression could be evolutionary tools to achieve the impressive variety of facial cartilage shapes in humans - a basis for facial individuality. Indeed, much attention has been focused on the role of enhancers in craniofacial evolution. Recently, by applying a combination of morphometry, molecular biology and mouse genetics, Attanasio et al. described numerous enhancers that are differentially active and take part in the development of a facial shape (*Attanasio et al., 2013*). In line with this, mutations in the enhancers that control the expression of *Fgf8*, another cartilage-inducing factor (*Abzhanov and Tabin, 2004*), also result in geometrical abnormalities of the nasal capsule (*Marinić et al., 2013*). It seems that many genes and pathways are involved in shaping the face (*Young et al., 2014*; *Hu et al., 2015a*; *Foppiano et al., 2007*; *Hu et al., 2015b*). This is not surprising, since the facial shaping includes many stages that are pre-chondrogenic, chondrogenic (including induction, growth, remodeling of the cartilage) and osteogenic. Processes of isotropic and anisotropic growth of the skeletal structures also play important roles in achieving the final geometry of the facial region (*Kaucka et al., 2017*). Despite such complexity and the enormous degree of spatio-temporal integration, the initial induction of cartilage guided by the brain and olfactory epithelium represents a key moment of facial skeleton formation. It may well also be an evolutionary

substrate driving the diversity of faces and snouts. Consequently, the fine-tuning of patterning and cartilage-inducing signals in neurosensory structures deserves further attention, including explorations of the diversity of corresponding genetic regulatory regions in human and animal genomes.

# Materials and methods

## Key resources table

| Reagent type species or resource | Designation | Source or reference | Identifiers | Additional information |
|---|---|---|---|---|
| Strain | Col2a1-CreERT2 | *Nakamura et al. (2006)* | | Received from S. Mackem |
| Strain | R26Confetti | https://www.jax.org/strain/013731 | | Received from H. Clevers |
| Strain | R26DTA | *Voehringer et al. (2008)* | | Received from Jackson |
| Strain | Six1 KO | *Grifone et al. (2005)* | | Received from P. Maire |
| Strain | Six1/4 double KO | *Grifone et al. (2005)* | | Received from P. Maire |
| Strain | Nkx2.2-Cre/Shhflx/flx | *Yu et al., 2013* | | Received from M. Matise |
| Strain | Ascl1 (Mash1) | *Cau et al. (1997)* | | Received from U. Marklund |
| Strain | B6.Cg-Shhtm1EGFP/creCjt/J | *Harfe et al. (2004)* | | Received from M. Hovorakova |
| Strain | Gli1-CreERT2 | https://www.jax.org/strain/007913 | | Received from M.Kasper |
| Strain | Gli1-lacZ | https://www.jax.org/strain/008211 | | Received from M.Kasper |
| Strain | Shh-GFP | https://www.jax.org/strain/008466 (*Chamberlain et al., 2008*) | | Received from Jackson |
| Strain | TgGBS-GFP | *Balaskas et al. (2012)* | | Received from A. Kicheva |
| Strain | Rosa-CAG-LSL-tdTomato-WPRE | https://www.jax.org/strain/007914 | | Received from M. Kasper |
| Strain | INV-500-C1 | F. Spitz (*Symmons et al., 2016*) | | Received from F. Spitz |
| Strain | INV6b-C2 | F. Spitz (*Symmons et al., 2016*) | | Received from F. Spitz |
| Strain | Del-90-C2 | F. Spitz (*Symmons et al., 2016*) | | Received from F. Spitz |
| Strain | DELC1-Z | F. Spitz (*Symmons et al., 2016*) | | Received from F. Spitz |
| Strain | DEL5-8 | F. Spitz (*Symmons et al., 2016*) | | Received from F. Spitz |
| Antibody | SOX9 | Sigma Aldrich, HPA001758 | | one to 1000 in PBS-T over night at RT |
| Antibody | ERBB3 | RnD Systems, AF4518 | | one to 500 in PBS-T over night at RT |
| Drug | Vismodegib | *LoRusso et al. (2011)* | | 0.1 mg/kg |
| Software | IMARIS | http://www.bitplane.com/ | | |
| Software | GOM Inspect | https://www.gom.com/de/3d-software/gom-inspect.html | | |
| Software | VGStudio Max | https://www.volumegraphics.com/en/products/vgstudio-max.html | | |

*Continued on next page*

Continued

| Reagent type species or resource | Designation | Source or reference | Identifiers | Additional information |
|---|---|---|---|---|
| RNAscope probes | Gli1 (311001), Gli2 (405771), Gli3 (445511), Smo (318411), Ptch1 (402811) and Ptch2 (435131) | https://acdbio.com/rnascope%C2%AE-technology-novel-rna-situ-hybridization-research-platform | | |

## Mouse strains and animal information

All animal work were approved and permitted by the Local Ethical Committee on Animal Experiments (North Stockholm Animal Ethics Committee) and conducted according to The Swedish Animal Agency´s Provisions and Guidelines for Animal Experimentation recommendations. Genetic tracing mouse strain *Nkx2.2-Cre* was described previously (*Yu et al., 2013*). *Col2a1-CreERT2* (*Ozaki et al., 2001*) (obtained from the laboratory of S. Mackem, NIH) strains (*Nakamura et al., 2006*) were coupled to *R26Confetti* mice that were received from the laboratory of Professor H. Clevers (*Snippert et al., 2010*). DTA strain (*Voehringer et al., 2008*) (*B6.129P2-Gt(ROSA)26Sortm1(DTA) Lky/J*, The Jackson Laboratory) was coupled to *Col2a1-CreERT2*. *Six1* and *Six1/4* double KO embryos were generated as described already (*Grifone et al., 2005*; *Laclef et al., 2003*). *Nkx2.2-Cre/Shh^{flx/flx}* embryos were received from the laboratory of Michael Matise. *Ascl1 (Mash1)* KO embryos were received from the laboratory of Ulrika Marklund. *B6.Cg-Shhtm1(EGFP/cre)Cjt/J* (*Harfe et al., 2004*) embryos were received from the laboratory of Maria Hovorakova (CAS). *Gli1-CreERT2* and *Gli1-lacZ* strains were obtained from the laboratory of Maria Kasper (Karolinska Institutet). *Gli1-CreERT2* was coupled with *R26Confetti* and *R26Tomato*.

The following strains were previously described: *Tg(GBS-GFP)* (*Balaskas et al., 2012*), *Shh-GFP* (JAX stock #008466 (*Chamberlain et al., 2008*). *Shh^{Hypo}* embryos are homozygous for *Shh-GFP* and their morphological phenotypes are not affected by the presence or absence of the *Tg(GBS-GFP)* transgene. Strains were bred and maintained on 129/Sv background, in accordance with license BMWFW-66.018/0006-WF/V/3b/2016 granted by the Austrian BMWFW.

*Gli1-LacZ* (https://www.jax.org/strain/008211), *Gli1-CreERT2* (https://www.jax.org/strain/007913) and *Rosa-CAG-LSL-tdTomato-WPRE* (https://www.jax.org/strain/007914) were used under the ethical permit: number S40/13, granted by South Stockholm Animal Ethics Committee.

Mice of the relevant genotype were mated overnight, and noon of the day of the plug was considered as E0.5. To induce genetic recombination of adequate efficiency, pregnant females of relevant couplings were injected intraperitoneally with tamoxifen (Sigma T5648) dissolved in corn oil (Sigma C8267). Tamoxifen concentration ranged from 1.5 to 5.0 mg per animal to obtain a range of recombination efficiency. Mice were sacrificed with isoflurane (Baxter KDG9623) overdose or cervical dislocation, and embryos were dissected out and collected into ice-cold PBS. Subsequently, the samples were placed into freshly prepared 4% paraformaldehyde (PFA) and depending on the developmental stage and the application they were fixed for 3–24 hr at 4°C on a roller. Subsequently, for the purpose of microscopy analysis, the embryos were cryopreserved in 30% sucrose (VWR C27480) overnight at 4°C, embedded in OCT media (HistoLab 45830) and cut into 18 µm to 30 µm sections on a cryostat (Microm). Embryos designated for CT analysis were then stained according to the protocol described beneath.

## Inhibition of hedgehog signaling

In order to inhibit SHH during embryonic development (stages E11.5 to E13.5), the pharmacological inhibitor Vismodegib (*LoRusso et al., 2011*) was injected intraperitoneally at a dosage of 0.1 mg per g of bodyweight of the pregnant mouse. Embryos were collected at E15.5 and fixed in 4% formaldehyde in PBS solution for 24 hr at +4°C with slow rotation.

## Histological staining

Slides were stained for mineral deposition using von Kossa calcium staining: 5% silver nitrate solution was added to the sections at a room temperature and exposed to strong light for 30 min. After that the silver nitrate solution was removed, and slides were washed with distilled water for 3 times during 2 min. 2.5% sodium thiosulphate solution (w/v) was added to the sections and incubated for five

mins. Slides were again rinsed for 3 times during 2 min in distilled water. The sections were then counterstained using Alcian blue. Alcian blue solution (0.1% alcian blue 8GX (w/v) in 0.1 M HCl) was added to the tissue for 3 min at room temperature and then rinsed for 3 times during 2 min in distilled water. Slides were then transferred rapidly into incrementally increasing ethanol concentrations (20%, 40%, 80%, 100%) and incubated in 100% ethanol for 2 min. Finally, the slides were incubated in two xylene baths (for 2 min and then for 5 min) before mounting and analysis.

## Immunohistochemistry, histological staining and EdU analysis

Frozen samples were sectioned at 18–30 µm depending on specific experiment. If needed, sections were stored at −20°C after drying 1 hr at room temperature, or processed immediately after sectioning. Primary antibodies used were: chicken anti-GFP (Abcam, 1:500, ab13970), rabbit anti-SOX9 (Sigma Aldrich, 1:1000, HPA001758), sheep anti-ERBB3 (RnD Systems, 1:500, AF4518). For detection of above-mentioned primary antibodies, we utilized 405, 488, 555 or 647-conjugated Alexa secondary antibodies produced in donkey (Invitrogen, 1:1000). Slices were mounted with 87% glycerol mounting media (Merck).

## Fluorescent in situ hybridization (RNAscope)

E12.5 and E13.5 embryos were collected, embedded immediately in OCT and snap frozen on dry ice. Tissue blocks were stored at −20°C until further use. 8-µm-thick cryosections were collected on Superfrost Plus slides and stored at −20°C until further use. Fluorescent in situ hybridization was performed for the genes *Gli1* (311001), *Gli2* (405771), *Gli3* (445511), *Smo* (318411), *Ptch1* (402811) and *Ptch2* (435131) using the RNAscope 2.0 Assay, reagents and probes according to manufacturer's instructions (*Wang et al., 2012*). RNAscope probes were designed commercially by the manufacturer and are available from Advanced Cell Diagnostics, Inc. being protected by patent.

## X-gal staining

E11.5, E12.5 and E13.5 embryos were fixed in 4% formaldehyde in PBS solution for 2–3 hr at +4°C with slow rotation. Following washes with PBS, embryos were incubated in X-gal staining solution (1 mg/ml X-gal; 2 mM MgCl2; 0.5 M potassium ferrocyanide; 0.5 M potassium ferricyanide in PBS) at 37°C, overnight, with gentle agitation. Samples were washed twice, 20 min each time at room temperature in PBS and imaged whole mount. When necessary, we proceeded to cryoprotection in 30% sucrose in PBS and embedding in OCT medium.

## Microscopy, volume rendering, image analysis and quantifications

Confocal microscopy was performed using Zeiss LSM880Airyscan CLSM instruments. The settings for the imaging of Confetti fluorescent proteins were previously described (*Snippert et al., 2010*). Image analysis has been performed using IMARIS Software (Bitplane, Zurich, Switzerland). Before performing manual segmentations of cartilages and mesenchymal chondrogenic condensations on all representative samples, we assessed the phenotype and the stability of the phenotype using analysis of multiple embryos (typically 3–5 per condition) on histological sections as well as whole-mount assessments of facial morphology and including usage of tomographic slices. In case the phenotype was stable, the representative embryos underwent 3D segmentation process, otherwise we manually segmented facial cartilage and bone from all experimental embryos (in case of *Col2a1-CreERT2/R26DTA* or *Shh^Hypo^*, please see *Figure 1—figure supplement 1* and *Figure 6—figure supplement 3*). For embryonic day E17.5 *Col2a1-CreERT2/R26DTA,* we performed segmentations of the most affected embryo from the litter (*Figure 1I–P* and graph in R). Other litter mates were analyzed using cryo-sections only (*Figure 1E–H*). Since we did not investigate fine differences in shape of the nasal capsule and rather concentrated on missing structures (septum or nasal capsule roof), we did not analyze fine shape differences morphometrically in a quantitative way. In special cases, where relevant, we utilized shape fitting analysis using GOM Inspect tool. We did not use any special randomization or masking of embryos during experimental and control group allocations.

## Tissue contrasting for µ-CT scanning

Staining protocol has been adapted and modified from the original protocol developed by Brian Metscher laboratory (*Metscher, 2009*). After embryo dissection in ice-cold PBS, the samples were

fixed in 4% formaldehyde in PBS solution for 24 hr at +4°C with slow rotation. Subsequently, samples were dehydrated in incrementally increasing ethanol concentrations (30%, 50%, 70%), 1 day in each concentration to minimize the shrinkage of the tissue. Samples were transferred, depending on the embryonic stage, into 1.0–1.5% PTA (phospho-tungstic acid) in 90% methanol for tissue contrasting. The PTA-methanol solution was changed every 2–3 days. E12.5 embryos were stained for 7 days, E15.5 embryos for 3 weeks and E18.5 embryos for 7 weeks. The contrasting procedure was followed by rehydration of the samples by incubation in ethanol series (90%, 70%, 50% and 30%) and shipped to the CT-laboratory for scanning. There the rehydrated embryos were embedded in 0.5% agarose gel (A5304, Sigma-Aldrich) and placed in polypropylene conical tubes (0.5, 1.5 or 15 ml depending on the sample size to minimize the amount of medium surrounding it) and to avoid the movement artifacts during X-ray computer tomography scanning.

## μ-CT analysis (micro computed tomography analysis) and 3D analysis

The μ-CT analysis of the embryos was conducted using the laboratory system GE phoenix v|tome|x L 240 (GE Sensing and Inspection Technologies GmbH, Germany), equipped with a 180 kV/15W maximum power nanofocus X-ray tube and flat panel detector DXR250 with 2048 × 2048 pixel, 200 × 200 μm pixel size. The embryos were fixed in polyimide tubes by 1% agarose gel to prevent tomographic movement artifacts. The exposure time of the detector was 900 ms in every of 2000 positions. Three projections were acquired and averaged for reduction of the noise in μ-CT data. The utilized power of the tube was 11 W given by acceleration voltage of 60 kV and tube current of 200 μA. X-ray spectrum was filtered by 0.1 mm of aluminium plate. The voxel size of obtained volumes (depending on a size of an embryo head) appeared in the range of 5 μm - 7 μm. The tomographic reconstructions were performed using GE phoenix datos|x 2.0 3D computed tomography software (GE Sensing and Inspection Technologies GmbH, Germany). The cartilage in the embryo head was segmented by an operator with semi-automatic tools within Avizo - 3D image data processing software (FEI, USA). The 3D segmented region was transformed to a polygonal mesh as a STL file. The mesh of the embryo head was imported to VG Studio MAX 2.2 software (Volume Graphics GmbH, Germany) for consequent modification of the mesh, like a surface smoothing, and 3D visualization. The software GOM Inspect V8 (GOM, Braunschweig, Germany) was implemented for comparisons of full shapes of the head. The triangular meshes of the surface of the heads represented by STL models were imported into the software, aligned and compared with parameters of maximum searching distance 1 mm and maximum opening angle 30°. All raw STL files are freely accessible via the following Dryad link: https://doi.org/10.5061/dryad.f1s76f2

The STL format can be opened with Paint 3D or Print 3D software.

## Light sheet microscopy and sample clearing

Whole heads from *Shh-GFP* embryos at E11.5 E12.5, E13.5 and E14.5 were cleared using a modified CUBIC protocol (*Susaki et al., 2014*). In brief, embryos were fixed by using 4% PFA in PBS for 4 hr at four degrees before incubating in CUBIC one solution (25% urea, 25% N,N,N′,N′-tetrakis-(2-hydroxypropyl) ethylenediamine and 15% Triton X-100) at 37°C under shaking conditions for 3 days. Subsequently, the samples were washed in PBS at RT. Next, samples were immersed in CUBIC two solution (50% sucrose, 25% urea, 10% 2,2′,2″-nitrilotriethanol, and 0.1% Triton X-100) and left shaking at RT for an additional 2–3 days before image acquisition.

Whole embryo head (E11.5–E14.5) GFP fluorescence images were acquired on a Light sheet Z.1 microscope (Zeiss) using a × 5 (EC Plan Neofluar 5×/0.16) detection objective,×5/0.1 illumination optics, and laser excitation at 488 nm. Samples were imaged in CUBIC two solution with a measured refractory index of 1.45. Each plane was illuminated from a single side of the sample. Whole images were obtained through tile scanning. 3D-rendered images were visualized with Arivis Vision4D for Zeiss (v. 2.11) or Imaris (v. 7.4.2, Bitplane).

Bitplane IMARIS software was subsequently used for 3D visualization and analysis of the light sheet tiles. By using the surface option in IMARIS the different parts of *Shh-GFP* have been highlighted.

## Acknowledgements

We would like to thank to Prof. Susan Mackem for the *Col2a1-CreERT2* mouse strain. MKau was supported by the SSMF fellowship (Svenska sällskapet för medicinsk forskning). IA was supported by Bertil Hållsten Research Foundation, Åke Wiberg Foundation, Vetenskapsrådet (VR), ERC Consolidator grant (STEMMING-FROM-NERVE, Project ID: 647844; ERC-2014-CoG), EMBO Young Investigator Program and Karolinska Institutet. JP was supported by a VR grant. KF was supported by VR MT, JK and TZ have been financially supported by the Ministry of Education, Youth and Sports of the Czech Republic under the project CEITEC 2020 (LQ1601) and by the project CEITEC Nano Research Infrastructure (MEYS CR, 2016–2019). MKas was supported by Cancerfonden, Swedish Foundation for Strategic Research and Ragnar Söderberg Foundation. KA was supported by Karolinska Institutet. FS lab was supported by EMBL. MEK was supported by the NovoNordisk Fonden Postdoctoral Stipend (Ref No NNF17OC0026874). MH was supported by Grant Agency of Czech Republic (14–37368G). Work in the AK lab is supported by the European Research Council under European Union Horizon 2020 research and innovation program grant 680037.

## Additional information

### Funding

| Funder | Grant reference number | Author |
| --- | --- | --- |
| Vetenskapsrådet | | Julian Petersen<br>Andrei S Chagin<br>Igor Adameyko |
| Svenska Sällskapet för Medicinsk Forskning | | Marketa Kaucka |
| Bertil Hållstens Forskningsstiftelse | | Igor Adameyko |
| Åke Wibergs Stiftelse | | Igor Adameyko |
| Karolinska Institutet | | Ulrika Marklund<br>Andrei S Chagin<br>Kaj Fried<br>Igor Adameyko |
| Ministerstvo Vnitra České Republiky | | Marketa Tesarova<br>Tomas Zikmund |
| Central European Institute of Technology | | Marketa Tesarova<br>Tomas Zikmund |
| Grantová Agentura České Republiky | | Maria Hovorakova |
| H2020 European Research Council | 680037 | Anna Kicheva |

The funders had no role in study design, data collection and interpretation, or the decision to submit the work for publication.

### Author contributions

Marketa Kaucka, Julian Petersen, Conceptualization, Data curation, Supervision, Validation, Investigation, Visualization, Methodology, Writing—original draft, Writing—review and editing; Marketa Tesarova, Data curation, Supervision, Investigation, Visualization, Methodology, Writing—review and editing; Bara Szarowska, Maria Eleni Kastriti, Jozef Kaiser, Resources, Data curation, Supervision, Methodology, Writing—review and editing; Meng Xie, Data curation, Writing—review and editing; Anna Kicheva, Conceptualization, Resources, Data curation, Supervision, Writing—original draft, Writing—review and editing; Karl Annusver, Data curation, Investigation, Methodology, Writing—review and editing; Maria Kasper, Resources, Data curation, Supervision, Methodology, Writing—original draft, Writing—review and editing; Orsolya Symmons, Resources, Data curation, Investigation; Leslie Pan, Resources, Data curation, Investigation, Writing—original draft, Writing—review and editing; Francois Spitz, Data curation, Supervision, Methodology, Writing—review and editing; Maria

Hovorakova, Conceptualization, Resources, Data curation, Supervision, Investigation, Writing—review and editing; Tomas Zikmund, Resources, Investigation, Methodology, Writing—review and editing; Kazunori Sunadome, Resources, Data curation, Investigation, Methodology, Writing—original draft, Writing—review and editing; Michael P Matise, Conceptualization, Resources, Data curation, Investigation, Methodology, Writing—original draft, Writing—review and editing; Hui Wang, Pascal Maire, Conceptualization, Resources, Data curation, Investigation, Methodology, Writing—review and editing; Ulrika Marklund, Hind Abdo, Patrik Ernfors, Andrei S Chagin, Kaj Fried, Igor Adameyko, Conceptualization, Resources, Data curation, Formal analysis, Supervision, Funding acquisition, Investigation, Methodology, Writing—original draft, Writing—review and editing; Maud Wurmser, Conceptualization, Resources, Data curation, Supervision, Investigation, Methodology, Writing—original draft, Writing—review and editing

### Author ORCIDs
Marketa Kaucka (iD) https://orcid.org/0000-0002-8781-9769
Julian Petersen (iD) http://orcid.org/0000-0002-7444-0610
Ulrika Marklund (iD) https://orcid.org/0000-0003-1426-1271
Andrei S Chagin (iD) https://orcid.org/0000-0002-2696-5850
Igor Adameyko (iD) https://orcid.org/0000-0001-5471-0356

### Ethics
Animal experimentation: All animal work was approved and permitted by the Local Ethical Committee on Animal Experiments (North Stockholm Animal Ethics Committee) and conducted according to The Swedish Animal Agency´s Provisions and Guidelines for Animal Experimentation recommendations. Permit numbers S40/13 and N226/15, granted by South Stockholm Animal Ethics Committee. The part, which was done in Austria at the Medical University of Vienna and IST was performed in accordance with license BMWFW-66.018/0006-WF/V/3b/2016 and BMWFW-66.009/0163-WF/V/3b/2016 granted by the Austrian BMWFW.

### Decision letter and Author response
Decision letter https://doi.org/10.7554/eLife.34465.027
Author response https://doi.org/10.7554/eLife.34465.028

## Additional files
### Supplementary files
• Transparent reporting form
DOI: https://doi.org/10.7554/eLife.34465.023

### Data availability
All data obtained including tomographic reconstructions will be freely available upon request since some datasets are considerably large (1TB and more) and depositing the full data is unfeasible. We have made a subset of the datasets available on the Dryad Digital Repository (http://dx.doi.org/10.5061/dryad.f1s76f2).

The following dataset was generated:

| Author(s) | Year | Dataset title | Dataset URL | Database, license, and accessibility information |
|---|---|---|---|---|
| Kaucka M, Petersen J, Tesarova M, Szarowska B, Kastriti ME, Xie M, Kicheva A, Annusver K, Kasper M, Symmons O, Pan L, Spitz F, Kaiser J, Hovorakova M, Zik- | 2018 | Data from: Signals from the brain and olfactory epithelium control shaping of the mammalian nasal capsule cartilage | http://dx.doi.org/10.5061/dryad.f1s76f2 | Available at Dryad Digital Repository under a CC0 Public Domain Dedication |

mund T, Sunadome K, Matise MP, Wang H, Marklund U, Abdo H, Ernfors P, Maire P, Wurmser M, Chagin AS, Fried K, Adameyko I

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
