## [Decision Letter]

Thank you for sending your article entitled "Signals from the brain and olfactory epithelium control shaping of the mammalian facial cartilage" for peer review at *eLife*. Your article has been evaluated by three peer reviewers, and the evaluation has been overseen by a Reviewing Editor and Marianne Bronner as the Senior Editor.

Major points:

1) *Nkx2.2* ablation experiments

The genetic ablation of SHH in the floor plate with the *Nkx2.2* driver is perhaps the most elegant experiment in the paper, resulting in a clear phenotype that includes truncation of the nasal capsule and other craniofacial defects. However, investigating the developmental basis of the defect (by examining earlier stages, looking at cell death and proliferation, etc.) could clarify the role of CNS SHH in the patterning of the nasal capsule. Furthermore, the experiment by itself may not be sufficient to confirm direct regulation of mesenchymal condensations by SHH. The authors should examine *Shh* signaling activity in the nasal capsule either through the use of *Gli1-LacZ* reporter mice or by analyzing the expression of *Shh* pathway readouts (*Ptch1, Ptch2, Smo, Gli1*).

2) *Col2a-DTA* ablation experiments

The authors should define what they mean by "mild chondrocyte ablation". It would be helpful to have a more quantitative and temporal understanding of how many cells are killed after the E12.5 induction in the DTA treatment and at what stage they are killed (as the difference between E13.5 to E15.5 can be dramatic). This would provide us with a better understanding of the precise stages at which the supposed SHH signal is required.

3) A timeline of the genetic perturbations

It would be helpful to consolidate all the data with a timeline scheme depicting the steps in facial cartilage formation and place all the different genetic variants onto this scheme – this would need to include the time point at which each set of mutants is first have an effect. For example, the *Nkx2.2* driver comes on much earlier than E12.5, and we know that there will be overall changes to the neuroepithelial tissues in this context. Similarly, the other mutants such as *Six1/Six4* are going to have major pathological effects much earlier than the cartilage condensation phases. Therefore, it seems difficult to eliminate the possibility that there is an earlier effect of SHH loss (e.g. changes in patterning or availability of mesenchymal tissues prior to cartilage condensation. This will of course raise the question of what is happening in the other mutants from E8+ through E12.5, but worth discussing.

4) *Shh* expressing cells in the olfactory epithelium

The experiment involving the ablation of the olfactory epithelium is difficult to interpret given that the whole tissue is being targeted. Thus it is possible that other signals coming from these cells are required for the proper patterning of the nasal capsule. Furthermore, *Six1/Six4* are expressed in the neural crest lineage and the head mesenchyme at earlier stages of development and thus we cannot rule out that the effects observed are indeed cell autonomous. This should be addressed.

5) Enhancer experiment

The enhancer experiments are tantalizing but very preliminary. First, the authors should clarify where these enhancers were published and what expression domains they produce. Have they been published (I could not locate a reference)? Furthermore, they would have to show how enhancer deletions results in changes of SHH expression. Unless the authors bridge the enhancer experiments with the previous data showing regional regulation of the nasal capsule, this section seems disconnected from the rest of the manuscript.

6) The Discussion does the data a great disservice. There is a wealth of information in these animal models that is glossed over, in part because not enough detail is given for us to understand the different mutants (especially *Shh* enhancers). Furthermore, because there is never a direct comparison between the different lines, it is difficult to interpret the data in total. I think this is crucial because effectively, the conclusions are drawn upon a series of correlations – if this is the bulk of the data, it will help the reader if all of the correlations are laid out in one unified vision at the end. Instead, the authors discuss some thoughts rationalising the evolution of the structures, which is rather a distraction from the findings. I don't see that it is necessary to make any evolutionary arguments; instead, I would like a bit more scholarship and context in the Discussion and some thought toward future directions.

---

## [Author Response]

Major points:1) Nkx2.2 ablation experimentsThe genetic ablation of SHH in the floor plate with the Nkx2.2 driver is perhaps the most elegant experiment in the paper, resulting in a clear phenotype that includes truncation of the nasal capsule and other craniofacial defects. However, investigating the developmental basis of the defect (by examining earlier stages, looking at cell death and proliferation, etc.) could clarify the role of CNS SHH in the patterning of the nasal capsule. Furthermore the experiment by itself may not be sufficient to confirm direct regulation of mesenchymal condensations by SHH. The authors should examine Shh signaling activity in the nasal capsule either through the use of Gli1-LacZ reporter mice or by analyzing the expression of Shh pathway readouts (Ptch1, Ptch2, Smo, Gli1).

We followed the advice of reviewers and explored if the effect of *Shh* on chondrogenesis is direct or indirect and if it coincides with chondrogenic stages in the face. For this we performed a number of experiments including:

1) The analysis of expression of SHH-pathway members using RNAscope (*Ptch1, Ptch2, Smo, Gli1, Gli2, Gli3*) and *Gli1-LacZ* reporter animals.

2) Lineage tracing with *Gli1-CreERT2* coupled to *R26Tomato* and *R26Confetti* reporters.

3) Analysis of SHH-pathway activity in mesenchymal chondrogenic condensations and cartilage by utilizing reporter *GBS-GFP* animals.

4) Injections of SHH-inhibitor vismodegib at chondrogenic stages (E11.5-E13.5) and comparison of the phenotype with “non-inducible” *Shh hypomorphs (Shh^Hypo^*).

Altogether, the results of these experiments formed three new main figures (Figure 5, Figure 6 and Figure 7) as well as a number of corresponding supplementary figures (Figure 5—figure supplement 1, Figure 6—figure supplement 1, Figure 6—figure supplement 2 and Figure 6—figure supplement 3). These data demonstrated that SHH plays a pre-chondrogenic and, most likely, patterning role that is not directly involved in facial cartilage differentiation.

In the initial version of the manuscript we focused on the brain and olfactory epithelium as centers directing cartilage development, and did not claim that SHH has a direct and selective effect on the cartilage development in the face. With the new results that we incorporated in the revised version, we explicitly clarify that the time-window of *Shh* activity occurs prior to the chondrogenic stages. This new finding suggests that *Shh*-induced pattern or secondary signal is involved in cartilage differentiation – possibilities that can be investigated in future studies.

Here is how we describe new results in the text of the Results section:

“To check if SHH from the brain and presumably from the olfactory epithelium acts directly on facial mesenchyme inducing chondrogenic differentiation or during cartilage growth, we analyzed embryos carrying a SHH-activity reporter *GBS-GFP*^27^ at different developmental stages ranging from E9 to E14.5. […] These results are also consistent with the phenotype of *Six1/Six4* double knockout embryos at E12.5 (Figure 4), and corroborate the notion of an early pre-cartilage onset of the phenotype.”

2) Col2a-DTA ablation experimentsThe authors should define what they mean by "mild chondrocyte ablation". It would be helpful to have a more quantitative and temporal understanding of how many cells are killed after the E12.5 induction in the DTA treatment and at what stage they are killed (as the difference between E13.5 to E15.5 can be dramatic). This would provide us with a better understanding of the precise stages at which the supposed SHH signal is required.

We agree and we followed the advice of the reviewers and defined the proportion of cartilage loss in presented nasal capsules from the DTA animals. We also performed new experiments (collecting embryos at earlier stage for the analysis – this time E15.5) and analyzed them. For this, we quantified the surface and volume of nasal capsule cartilage in control and DTA embryos analyzed at E15.5 and E17.5. For E17.5, the loss of cartilage in DTA sample was mean 30,7% of the cartilage surface decrease at E15.5 and mean 35,2% at E17.5. The new results can be found in Figure 1 and Figure 1—figure supplement 1.

The cartilage of the nasal capsule is fully set for the first time only at E15.5 (before that different parts are generated from chondrogenic condensations, see our paper Kaucka et al., 2017), and, thus, in the new experiment with another scheme of treatment aimed at higher cartilage loss, we analyzed the nasal capsule at E15.5 and not at E17.5. The principal results were consistent with the original experiment and showed how bone development becomes affected. Despite the reduction of nasal capsule cartilage, the membranous bones were induced adjacently in a tight association with chondrocranial parts. Please see Figure 1P,Q,R.

3) A timeline of the genetic perturbationsIt would be helpful to consolidate all the data with a timeline scheme depicting the steps in facial cartilage formation and place all the different genetic variants onto this scheme – this would need to include the time point at which each set of mutants is first have an effect. For example, the Nkx2.2 driver comes on much earlier than E12.5, and we know that there will be overall changes to the neuroepithelial tissues in this context. Similarly, the other mutants such as Six1/Six4 are going to have major pathological effects much earlier than the cartilage condensation phases. Therefore, it seems difficult to eliminate the possibility that there is an earlier effect of SHH loss (e.g. changes in patterning or availability of mesenchymal tissues prior to cartilage condensation. This will of course raise the question of what is happening in the other mutants from E8+ through E12.5, but worth discussing.

We generated an illustration (Figure 7) that depicts the timeline of all experiments and the beginning of effects. We cross-compare the phenotypes in Results and also in Discussion, so there is a clarity of when the action of mutation takes place and how different conditions are similar to each other or different.

4) Shh expressing cells in the olfactory epitheliumThe experiment involving the ablation of the olfactory epithelium is difficult to interpret given that the whole tissue is being targeted. Thus it is possible that other signals coming from these cells are required for the proper patterning of the nasal capsule. Furthermore, Six1/Six4 are expressed in the neural crest lineage and the head mesenchyme at earlier stages of development and thus we cannot rule out that the effects observed are indeed cell autonomous. This should be addressed.

We agree and understand the reviewer’s concerns. This is indeed great advice, and we discussed this moment extensively. It might be good to mention that we never claimed that it is SHH from the olfactory epithelium that induces the nasal capsule roof. Targeting *Shh* exclusively in the nasal placode is currently impossible. In the revised version of the manuscript, we discussed this issue in a more clear way since we totally agree that other signals derived from olfactory placode in principle can guide the development of nasal capsule roof.

“The induction of such 3D-shapes is unlikely to be achieved by signals from a single site and might require more sources including spatially opposed brain and olfactory structures. […] This will require further investigations.”

We provided additionally new data by generating and checking the key early stage of *Six1/Six4* double knockout (E12.5) and also by exploring a single knockout of *Six1 (Six4* knockout showed no phenotype). Here is the new text that we have in a revised version of the manuscript:

“In addition to the expression in olfactory placodes, *Six1* and *Six4* are expressed in different parts of early facial mesenchyme (Kobayashi et a., 2007; Grifone et al., 2005. […] The phenotype in *Six1^-/-^* embryos mostly included a narrowing of the posterior nasal capsule with mild septal defects (Figure 4L,M and Figure 4—figure supplement 1E).”

“Complementary to the ablation of *Shh* by *Nkx2.2-Cre*, the loss of the olfactory epithelium in *Six1/Six4* double knockout mutants leads to the absence of the nasal capsule roof, while the nasal septum cartilage stays largely in place. […] Already at the earliest steps, they are laid down as highly complex 3D-geometries (Kaucka et al., 2017).”

5) Enhancer experimentThe enhancer experiments are tantalizing but very preliminary. First, the authors should clarify where these enhancers were published and what expression domains they produce. Have they been published (I could not locate a reference)?

This is clarified in the main text. The initial discovery of these genomic regulatory regions is described in a recent manuscript from Francois Spitz laboratory (Symmons et al., 2016).

Furthermore, they would have to show how enhancer deletions results in changes of SHH expression. Unless the authors bridge the enhancer experiments with the previous data showing regional regulation of the nasal capsule, this section seems disconnected from the rest of the manuscript.

We did our best to run more experiments and to show the plethora of the phenotypes that explain more clearly now how brain-specific *Shh* enhancers influence facial cartilage development. For the most critical mutations of *Shh* regulatory regions, we either report *Shh-LacZ* reporter data showing the shift of *Shh* expression in the forebrain or we cite the data of this kind obtained and published previously. In any case, we decided to move all data about enhancers to the supplementary information in this manuscript because of its accessory function.

Here are the major updated results also outlined in two supplementary figures (Figure 6—figure supplement 1 and Figure 6—figure supplement 2):

“Tissue-specific expression of *Shh* is known to be controlled by multiple enhancers. Some, which may regulate *Shh* expression in the cranial region, have been characterized (Yao et al., 2016; Jeong et al., 2006; Sagai et al., 2009). […] Taken together, local tuning of SHH expression by various enhancers (including brain-specific regulatory elements) seemingly controls discrete steps of chondrocranial patterning, which may represent a key evolutionary mechanism shaping animal snouts and faces.”

6) The Discussion does the data a great disservice. There is a wealth of information in these animal models that is glossed over, in part because not enough detail is given for us to understand the different mutants (especially Shh enhancers). Furthermore, because there is never a direct comparison between the different lines, it is difficult to interpret the data in total. I think this is crucial because effectively, the conclusions are drawn upon a series of correlations – if this is the bulk of the data, it will help the reader if all of the correlations are laid out in one unified vision at the end.

We expanded on the direct comparisons between the relevant mutant lines and we also added a new whole figure outlining the phenotypes and timing of effects (see Figure 7).

Instead, the authors discuss some thoughts rationalising the evolution of the structures, which is rather a distraction from the findings. I don't see that it is necessary to make any evolutionary arguments; instead, I would like a bit more scholarship and context in the Discussion and some thought toward future directions.

We agree and we expanded the discussion towards molecular mechanism of SHH action on chondrogenesis, as well as provided some additional lines about future directions. We also kept the evolutionary tilt in the Discussion text since we hope that our findings will be considered important for EvoDevo aspect of craniofacial research.